



# Lake thermal structure drives inter-annual variability in summer anoxia dynamics in a eutrophic lake over 37 years

Robert Ladwig[1], Paul C. Hanson[1], Hilary A. Dugan[1], Cayelan C. Carey[2], Yu Zhang[3], Lele Shu[4], Christopher J. Duffy[5], Kelly M. Cobourn[6]

[1]Center for Limnology, University of Wisconsin-Madison, Madison, WI, USA
[2]Department of Biological Sciences, Virginia Tech, Blacksburg, VA, USA
[3]Earth and Environmental Sciences, Los Alamos National Laboratory, Los Alamos, NM, USA
[4]Department of Land, Air and Water Resources, University of California, Davis, Davis, CA, USA
[5]Department of Civil & Environmental Engineering, The Pennsylvania State University, State College, PA, USA
[6]Department of Forest Resources and Environmental Conservation, Virginia Tech, Blacksburg, VA, USA

*Correspondence to*: Robert Ladwig (rladwig2@wisc.edu)

**Abstract.** The concentration of oxygen is fundamental to lake water quality and ecosystem functioning through its control over habitat availability for organisms, redox reactions, and recycling of organic material. In many eutrophic lakes, oxygen depletion in the bottom layer (hypolimnion) occurs annually during summer stratification. The temporal and spatial extent of summer hypolimnetic anoxia is determined by interactions between the lake and its external drivers (e.g., catchment characteristics/nutrient loads, meteorology), as well as internal feedback mechanisms (e.g., organic matter recycling, phytoplankton blooms). How these drivers interact to control the evolution of lake anoxia over decadal time scales will determine, in part, the future lake water quality. In this study, we used a vertical one-dimensional hydrodynamic-ecological model (GLM-AED2) coupled with a calibrated hydrological catchment model (PIHM-Lake) to simulate the thermal and water quality dynamics of the eutrophic Lake Mendota (USA) over a 37-year period. The calibration and validation of the lake model consisted of a global sensitivity evaluation as well as the application of an evolutionary optimization algorithm to improve the fit between observed and simulated data. By quantifying stability indices (Schmidt Stability, Birgean Work, stored internal heat), we identified spring mixing and summer stratification periods, and quantified the energy required for stratification and mixing. To qualify which external and internal factors were most important in driving the inter-annual variation in summer anoxia, we applied a random-forest classifier and multiple linear regression to modeled ecosystem variables (e.g., stratification onset and offset, ice duration, gross primary production.) Lake Mendota exhibited prolonged hypolimnetic anoxia each summer, lasting between 50-60 days. The summer heat budget, as well as the timing of thermal stratification, were the most important predictors of the spatial and temporal extent of summer anoxia periods in Lake Mendota. An earlier onset of thermal stratification in combination with a higher vertical stability strongly affected the duration and spatial extent of summer anoxia. As the heat budget depended primarily on external meteorological conditions, the spatial and temporal extent of summer anoxia in Lake Mendota is likely to increase in the near future as a result of projected climate change in the region.



## 1 Introduction

The availability of dissolved oxygen in lakes governs ecological habitats and niches, the rates of redox reactions, and the processing of organic matter throughout the water column (Cole and Weihe, 2016). When a lake is thermally stratified,

metabolism in the surface layer (epilimnion) can act as a net source or sink of oxygen, depending on the balance of gross primary production and ecosystem respiration, and deviations of dissolved oxygen from saturation values are modulated by atmospheric exchange (Odum, 1956). Additionally, entraining inflows can also act as an important oxygen sink or source depending on the lake morphometry, the inlet discharge, and the carrying capacity for dissolved oxygen (Burns, 1995). Below the thermocline, dissolved oxygen is depleted in the bottom layer (hypolimnion) by organic matter mineralization in the water

column and the sediment oxygen demand (Livingstone and Imboden, 1996). These oxygen depletion processes can be quantified as either a volumetric sink (e.g., due to organic matter mineralization in the water column) or as an areal sink (e.g. oxygen demand in the sediments). The depletion rates of oxygen depend on the organic matter pool (Müller et al., 2012, 2019), the trophic status of the lake (Rhodes et al., 2017; Rippey and McSorley, 2009), the area to volume relationship over depth (Livingstone and Imboden, 1996), and the chemical demand of the water column and sediments (Yin et al., 2011).

While the biogeochemistry of lake oxygen is well studied, there is much to be learned about decadal-scale controls over ecosystem patterns in oxygen and the interactions of external drivers with internal processes that control those patterns. Oxygen depletion in the hypolimnion that results in hypoxia (dissolved oxygen < 2 mg L$^{-1}$; Diaz and Rosenberg, 2008) and anoxia (dissolved oxygen < 1 mg L$^{-1}$;  Nürnberg, 1995b) is a product of interacting external drivers (e.g., climate, land use practices in the catchment) that control mass fluxes (Jenny et al., 2016b), morphometric characteristics, and productivity that

influences vertical transport and water column stability (Meding and Jackson, 2003). Unprecedented changes to the climate and catchment land use are likely to have nonlinear consequences on aquatic water bodies and will potentially intensify hypolimnetic anoxia (Jenny et al., 2014, 2016a; Sánchez-España et al., 2017).

The influence of physical controls on lake anoxia is of particular interest because it provides clues to how lakes might respond to long-term changes in exogenous drivers. The timing of anoxia has been found to be strongly related to the onset of

stratification, as well as sediment oxygen demand in small eutrophic lakes (Biddanda et al., 2018; Foley et al., 2012). A reduction of winter/spring mixing and increase in stratification can be major drivers of deep-water oxygen depletion (North et al., 2014). For Lake Mendota (USA), Snortheim et al. (2017) concluded that changes in air temperature was the main driver of the spatio-temporal extent of summer hypolimnetic anoxia. The question remains open, even for single ecosystems: Under what circumstances and at what time scales does thermal stratification strength act as the dominant driver of hypolimnetic

anoxia versus biogeochemical processes?

Studying decadal scale lake anoxia would benefit from an ecosystem-scale metric of lake anoxia and an analytical framework for tying that metric to physical and biological processes. Several metrics of oxygen availability have been developed by previous studies, such as mean areal hypolimnetic oxygen depletion (AHOD, in mg O$_2$ m$^{-2}$ d$^{-1}$, Cornett and Rigler, 1979; Hutchinson, 1938), volumetric rate of oxygen consumption (VOD, in mg O$_2$ m$^{-3}$ d$^{-1}$, Cornett, 1989), areal





hypolimnetic mineralization (AHM, in g $O_2$ $m^{-2}$ $d^{-1}$, Matzinger et al., 2010), and the Anoxic Factor (AF, in days, Nürnberg, 1995a, 1995b, 2004). Compared to the Anoxic Factor, most metrics calculate an oxygen depletion rate, whereas the Anoxic Factor provides an integrated metric that includes the spatial as well as temporal dimensions of anoxia per season. As the Anoxic Factor sums up the product of anoxia duration with the corresponding area, it is therefore a useful metric to evaluate long-term dynamics of hypolimnetic anoxia, and to compare the intensity of anoxia between years and different study sites.

The Anoxic Factor and its derivative the Hypoxic Factor (the difference being the threshold of dissolved oxygen; (Nürnberg, 2004) have been used in several studies, and observations range from 0 to 83 days per summer for different lake ecosystems (Nürnberg, 1995b).

Hydrodynamic-water quality models are an established approach to studying lake physical and biological responses to external drivers (Hipsey et al. 2019). An advantage of a lake ecosystem model calibrated to observed long-term data is that

it can reproduce finer temporal and spatial resolution than observational data permits for most ecosystems (Stanley et al., 2019) allowing for the investigation of complex ecosystem dynamics (Ward et al., 2020). By applying an ecosystem model driven by sub-daily meteorological and daily hydrological inflow data, physical processes relevant to hypolimnetic oxygen depletion (such as the onset and seasonal evolution of thermal stratification and gas transfer velocities) can be resolved at an hourly resolution, and can subsequently be incorporated into stochastic models to gain an understanding about the relationships

between drivers and their respective impacts on hypolimnetic anoxia (Snortheim et al., 2017). Results from deterministic lakes models can be analysed using statistical models to derive general empirical relationships. Results can also be compared with alternative, deductive approaches, which tend to be simpler models meant to reproduce gross ecosystem properties. An example relevant to lake anoxia is the simple deductive hypolimnetic oxygen depletion model by Livingstone and Imboden (1996), which established that already minor year-to-year meteorological variations during spring can cause an expansion of

the thickness of the summer anoxia layer.

This study aims to determine the extent to which physical, chemical, and biological (internal and external) factors control the inter-annual variability of the summer Anoxic Factor over 37-years in the eutrophic Lake Mendota. We use a lake hydrodynamic-water quality model to generate fine scale ecosystem states and fluxes based on observational data from the North Temperate Lakes Long Term Ecological Research program. Further, we use a deductive lake anoxia model and data-

driven empirical models to evaluate observed and simulated data and to determine broad-scale control over lake anoxia. We answer three questions with our modeling framework: (1) Overall, do internal biogeochemical processes or external loadings control year-to-year variability of the Anoxic Factor? (2) What are essential in-lake physical and biological controls over the long-term variability in anoxia? (3) As the timing of thermal stratification governs hypolimnetic oxygen depletion, what is the year-to-year variability of Lake Mendota's head budget? Answers to these questions will further our understanding of lake

ecosystem responses to climate or landscape changes and support future management decisions.



## 2 Materials and Methods

### 2.1 Study Site

Lake Mendota is a 39.61 km$^2$, 25 m deep, eutrophic lake in southern Wisconsin (USA). The lake has a mean water residence time of 4.3 years (McDonald and Lathrop, 2017). Lake Mendota's mixing regime is characterized by a summer stratification period from late April through October and an inverse winter stratification period under ice (Brock, 1985). Lake Mendota's air temperature ranges from -39 to 40 °C with a mean annual value of 8 °C, and an annual precipitation ranging from 540 to 990 mm with an average of 780 mm (Lathrop, 1992). The 604 km$^2$ watershed is dominated by agricultural land (67%) and developed urban land (22%) (Duffy et al., 2018). Since 1995, physical, chemical, and biological characteristics have been sampled biweekly to monthly by the North Temperate Lakes Long Term Ecological Research Program (Magnuson et al., 2006). We note that Lake Mendota is a "hard water" lake with pH > 7 and exhibits consistently high dissolved inorganic carbon concentrations, with speciation dominated by bicarbonate and carbonate.

### 2.2 Driver Data Acquisition

Meteorological forcing data were obtained from the second phase of the North American Land Data Assimilation System (NLDAS-2, Xia et al., 2012). The data from the grid cell were centered at -89.4375, 43.0625. The NLDAS-2 grid cells are 1/8th-degree spacing and data are at an hourly resolution from January 1, 1979 to present (Mitchell, 2004). Meteorological parameters used in this study included wind speed, air temperature, specific humidity, surface pressure, surface downward short- and longwave radiation, and total precipitation, which were used primarily as boundary data for GLM-AED2. Relative humidity was calculated *post hoc* as a function of specific humidity, air temperature, and surface pressure.

To quantify the water budget in Lake Mendota, we simulated the water inflow from the catchment (through stream flow, overland flow, and groundwater flow) to the lake, and water outflow from the lake to the catchments using a physically based distributed hydrologic model, PIHM-Lake (Penn State Integrated Hydrologic Model, Qu and Duffy, 2007). PIHM-Lake integrates hydrologic processes in a lake-catchment coupled system simulating the surface and subsurface hydrologic interaction within the catchment and between the catchment and the lake. Hydrologic interactions within the catchment are modeled in three dimensions, while the lake is represented in PIHM-Lake as a simplified one-dimensional bucket model assuming a uniform lake surface area and depth. PIHM-Lake tracks the change in water storage from the watershed's vegetation canopy, ground surface, unsaturated soil zone, saturated soil zone, and lake by using the semi-discrete finite volume method and a triangular irregular network (TIN). The PIHM-Lake simulation covers a 37-year period (from 1979 to 2015), and was calibrated and validated using in-situ measured stream flow, groundwater table level, lake surface water level, and lake outflow from the US Geological Survey.

Surface nutrient loadings from the Yahara River and Pheasant Branch inflows into Lake Mendota were estimated by regression models using discharge and nutrient concentration data from USGS gages. Combined with the simulated discharge time series from PIHM-Lake, these regressions were using to compute daily loading data. For a complete description of the





inflow loading regression see Weng et al. (2020). To provide information regarding adsorbed soluble reactive phosphate, we doubled measured total phosphorus (TP) concentrations and applied specific ratios to individual phosphorus forms (Farrell et

al., 2020; Snortheim et al., 2017; Weng et al., 2020). As direct measurements of inlet loadings of refractory organic matter, dissolved inorganic carbon (DIC) and silica were not available, we assumed constant average values for the inflow loadings similar to the long-term mean values of the water column.

Monitored NTL-LTER data from 1995 to 2015 were used for model verification. Data included water temperature and dissolved oxygen concentrations (Magnuson et al., 2020b) with a vertical spatial resolution of 1 m from the surface to 24

m. Data were measured biweekly during summer, monthly during fall and once per winter. The dissolved oxygen data set was complemented with historical measured dissolved oxygen data from 1992 to 1994. NTL-LTER data also included pH, dissolved inorganic carbon, dissolved organic carbon, nitrate, ammonia, soluble reactive phosphate and silica sampled at the depths 0, 3, 8, 10, 12, 14, 16, 18, 20 and 22 m (Magnuson et al., 2020a). Surface integrated samples of epilimnetic chlorophyll-a and Secchi depth were used to evaluate GLM-AED2's predictions of phytoplankton biomass and light extinction (Magnuson

et al., 2020c, 2020d).

## 2.3 Modeling Framework

Our modeling framework to investigate drivers of hypolimnetic anoxia consisted of three components (Figure 1):

(1) *Deductive model:* A deductive model formulated by Livingstone and Imboden (1996) was run on the monitored field data to characterize the empirical relationships between observed dissolved oxygen data and oxygen depletion processes

and to quantify the contributions of water column and sediments to hypolimnetic oxygen demand (Figure 1). The deductive model furthered our ecosystem-scale understanding of the partitioning between volumetric and areal oxygen depletion sinks in Lake Mendota. Therefore, this approach was used independently of the other modeling approaches as a "check" on the sediment oxygen demand rates of Lake Mendota used in GLM-AED2.

(2a) *GLM-AED2*: To gain a more mechanistic understanding of how processes driving oxygen depletion lead to

ecosystem-scale oxygen dynamics, we used the vertical one-dimensional hydrodynamic water quality model, GLM-AED2 (Hipsey et al., 2019b). GLM-AED2 uses meteorological, hydrological, and nutrient load data as inputs and predicts lake physical, chemical, and biological dynamics, including those of dissolved oxygen. The advantage of using GLM-AED2 is that it quantifies and tracks processes relevant to oxygen cycling using well-accepted physical and biogeochemical interactions that otherwise are difficult to infer from observational data alone (Figure 1). Although GLM-AED2 is a deterministic model,

hypolimnetic anoxia is an emergent ecosystem property that derives from a complex suite of interactions within the model (Snortheim et al., 2017). Therefore, we used GLM-AED2 to simulate and track states and fluxes of modelled variables.

(2b) *Regression model*: To derive generalized relationships between the interannual variation in hypolimnetic anoxia and the driving data, as well as the output from GLM-AED2, we used statistical models on our combined dataset of monitored and modeled data. Because the number of potential candidate predictors is high, we used a machine learning approach to





determine the most significant predictors of seasonal hypolimnetic anoxia at the inter-annual scale (Figure 1). These predictors were then used in a multiple linear regression to rank their influence on hypolimnetic anoxia.

## 2.4 Deductive Model

Using temporal and spatial linearly interpolated observed dissolved oxygen data, we applied the simple deductive oxygen depletion model according to Livingstone and Imboden (1996) in which the oxygen depletion rate $J(z)$ at depth $z$ is

$$J(z) = J_v(z) + J_A(z)\alpha(z), \tag{1}$$

where $J_V$ is the volume sink (mass per volume per time) representing organic matter mineralization processes, e.g. microbial respiration in the water column, $J_A$ is the area sink (mass per area per time) representing sediment oxygen demand, and $\alpha$ is a function for the ratio of sediment area to water volume (Bossard and Gächter, 1981; Livingstone and Imboden, 1996):

$$\alpha(z) = -\frac{1}{A(z)}\frac{dA(z)}{dz}. \tag{2}$$

We used observed dissolved oxygen data from 1992 to 2015 (measured biweekly after ice offset) to calculate the specific oxygen depletion over depth for each year individually from the date of spring mixing offset to the date when oxygen concentrations were below 2 mg L-1 (criterium for hypoxia). Only dissolved oxygen data below a depth of 15 m were used. The derivatives of area to depth were approximated by using forward and backward differencing. The terms $J_V$ and $J_A$ were assumed to be constant for every year (assuming the hypolimnion to be homothermic) and were determined by using weighted
linear regression.

## 2.5 GLM-AED2

For simulating Lake Mendota, we used the coupled 1D vertical hydrodynamic-ecological model GLM-AED2 (GLM: v.3.1.0a1, AED2: 1.3.4, developed by University of Western Australia, Hipsey et al. 2019). The hydrodynamic model GLM incorporates a flexible Lagrangian grid with each layer's thickness dynamically changing in response to the respective water
density (Hipsey et al., 2019b). Surface mixing processes are computed via an energy balance approach that compares the available (turbulent) kinetic energy to the internal potential energy of the water column (Hipsey et al., 2019b).

The water quality module, AED2, was configured to simulate the dynamics of dissolved oxygen, silica, inorganic carbon, organic matter (refractory, particulate and dissolved C, N and P), inorganic matter (refractory, particulate and dissolved C, N and P) as well as $PO_4$, $NO_3$, and $NH_4$, and two functional phytoplankton groups (representing diatoms and cyanobacteria,
Appendix Table A1). The model was run on an hourly time step and output data were saved at a daily timestep on noon. The thickness of each model layer (set to a maximum of 75 layers) could vary between 0.15 and 1.5 m with a minimum layer volume of 0.1 m³. The source code of the model's version, configuration files, input and output data are stored and accessible at Ladwig et al. (2020) via the Environmental Data Initiative.

A global sensitivity analysis (Morris Method after Morris 1991) was conducted to identify the most influential
parameters for the predictions of water temperature, dissolved oxygen, dissolved inorganic carbon, silica, nitrate and



phosphate, respectively. Using the Morris Method with 10 iterative runs, the distributions of the absolute elementary effects (the model change quantified by a fit function, here the root-mean squared error (RMSE) between observed and simulated data) of each parameter were calculated. According to Morris (1991) and Saltelli et al. (2004), the mean of the absolute elementary effects represents the overall sensitivity of the model outcome to each parameter, and the standard

deviations are a metric of the interactions between different parameters. All parameters with a normalized mean elementary effect over 0.1 were declared sensitive and were used for the calibration.

According to our calculated absolute elementary effects, we included the following 10 parameters in the calibration, listed according to their respective state parameter (Appendix Figure A1). Water temperature: bulk aerodynamic coefficient for sensible heat transfer (ch), long-wave radiation factor (lw_factor), mean sediment temperature (sed_temp_mean),

shortwave radiation factor (sw_factor). Dissolved oxygen: Sediment flux (Fsed_oxy), mineralization rate of dissolved organic matter (Rdom), temperature multiplier for sediment flux (theta_sed_oxy). Dissolved inorganic carbon: Sediment flux (Fsed_dic), half-saturation constant for oxygen dependence on sediment flux (Ksed_dic), temperature multiplier for sediment flux (theta_sed_dic). Silica: Sediment flux (Fsed_rsi), half-saturation constant for oxygen dependence on sediment flux (Ksed_rsi), temperature multiplier for sediment flux (theta_sed_rsi). Nitrate: Sediment flux (Fsed_nit), half-saturation constant

for oxygen dependence on denitrification (Kdenit), half-saturation constant for oxygen dependence on sediment flux (Ksed_nit), maximum reaction rate of denitrification at 20 °C (Rdenit). Phosphate: Sediment flux (Fsed_frp), half-saturation constant for oxygen dependence on sediment flux (Ksed_frp), temperature multiplier for sediment flux (theta_sed_frp).

We applied a combination of an automatic calibration technique and manual calibration for the calibration period from 2005 to 2015. First, the derivate-free, evolutionary optimization algorithm (CMA-ES, Hansen (2016)) was used to

minimize the RMSE between observed and simulated data (data were split into a calibration, 2005-2015, and a validation period, 1995-2004). We used a time period prior to the calibration period for validation to stress-test the model by applying it a time period with potential different ecological characteristics. The model parameters were calibrated iteratively (and fixed for the next calibration step) in the following order: water temperature, dissolved oxygen, dissolved inorganic carbon, silica, nitrate and, last, phosphate. We did not calibrate for phytoplankton functional group biomass because it was out of scope for

this analysis, but the model qualitatively recreated observed seasonal succession. Calibration of water temperature and dissolved oxygen concentrations were run for 300 iterations, the other variables for 200 iterations. The fit criteria were root-mean square error (RMSE), Nash-Sutcliffe Coefficient of Efficiency (NSE) and Kling-Gupta Efficiency (KGE) (Gupta et al., 2009) for the calibration period, the validation period and the total time period. The advantage of combining an automatic approach and a manual post-calibration for an overparameterized model such as GLM was that CMA-ES first limited the

possible parameter space of each parameter, then in a second calibration step, parameters could be manually changed to improve overall dynamics and behavior without relying on a fixed objective function. The manual calibration was done to ensure that the model was not overoptimized with unrealistic parameter combinations of the biological parameters. This calibration approach was done in accordance with other aquatic ecosystem modeling studies (Fenocchi et al., 2019; Mi et al., 2020), that did not apply computational optimization to water quality models.





## 2.6 Post-Processing of GLM-AED2 Output

We quantified two heat budget metrics from simulated water temperature data, the Schmidt Stability (Idso, 1973; Read et al., 2011; Schmidt, 1928) and the Birgean Work (Birge, 1916; Idso, 1973). Schmidt Stability ($St$) is a stability index that expresses the amount of energy needed to mix the entire water column to uniform temperatures without affecting the amount of internal energy, whereas Birgean Work ($B$) is a stability index that quantifies the amount of external energy that is theoretically needed to build up the current stratification from a hypothetical completely mixed state. The sum of both terms, the total work $G$, gives an estimate of the energy needed to keep a lake isothermal during stratified conditions:

$$G = St + B, \tag{3}$$

$$G = \frac{g}{A_s} \int_0^{z_m} A_z(1 - \rho_z)(z_v - z)dz + \frac{g}{A_s} \int_0^{z_m} A_z(1 - \rho_z)zdz, \tag{4}$$

where $g$ is gravity, $A_s$ is the surface area (m$^2$), $z_m$ is the maximum depth (m), $A_z$ is the respective area at the depth $z$, $\rho_z$ is the respective density at the depth $z$ (kg m$^{-3}$), $z_v$ is the depth of the center of volume ($z_v = \frac{1}{V}\int_0^{z_m} A_z z dz$), and $V$ is the volume (m$^3$). The stagnancy of deep water can be quantified by calculating a heat budget ratio ($HBR$):

$$HBR = \frac{G}{B}, \tag{5}$$

which compares the amount of energy needed to maintain isothermal conditions to the amount of available external energy (Kjensmo, 1994). Here, an increased stagnancy of deep waters results in a reduced exchange of fluxes between the surface mixed and the bottom layer. Therefore, HBR values > 1 indicate the isolation of the bottom water layers from surface fluxes in a lake.

Internal energy - as the stored thermal energy in the water column - was quantified using the R package rLakeAnalyzer (Winslow et al., 2019) as:

$$E_{internal} = \frac{1}{A_s} \int_0^{z_m} T_z * c_w * m_z dz, \tag{6}$$

where $T_z$ is the water temperature at depth z (°C), $c_w$ is the specific heat of water ($J\ kg^{-1}\ K^{-1}$), and $m_z$ is the mass of water at depth z ($kg$).

The thermocline depth was defined as a planar separation between the surface mixed and the bottom stagnant layer. The specific depth of this planar thermocline was quantified as the depth of the maximum density difference over the vertical axis where the minimum water temperature was above 4 °C and the density difference between surface and bottom layer was above 0.1 kg m$^{-3}$, signaling stratified conditions.

The temporal and spatial extent of anoxia during the summer season was quantified using the Anoxic Factor:

$$AF = \sum_{i=1}^{n} \frac{t_i A_i}{A_s}, \tag{7}$$


which sums the product of the anoxia duration $t$ (days) with the corresponding area $A$ (m²) to the total surface area $A_S$ when the in-water dissolved oxygen concentrations were below a threshold of 1 mg L⁻¹ (Nürnberg, 1995b). As the Anoxic and

Hypoxic Factor use the same equation with different thresholds relating essentially all anoxia information also to hypoxia, we focused on only quantifying the Anoxic Factor in this study. Anoxic Factor was calculated using the simulated dissolved oxygen concentrations as well as the approximately bi-weekly monitored field data, in which case data were temporally and spatially interpolated using an ensemble of approaches (linear, constant and spline interpolation between neighboring data points). We quantified the seasonal Anoxic Factor only during summer as Lake Mendota is not experiencing winter

hypolimnetic anoxia under ice.

## 2.7 Regression Model

We evaluated 22 candidate predictors on their relative importance in predicting the simulated summer Anoxic Factor of the respective year $n$ (see Table 1 for an overview and further explanation). For the calculation of certain candidate predictors, the water column was separated into an upper layer (from surface to a depth of 10 m) and a lower layer (from 10 m to maximum

depth). Although this is a rough approximation, this depth roughly represents the thermocline depth and further separates the water column into a zone without light limitation and one with light limitation. To represent external control processes, we included the seasonal total phosphorus inflow and seasonal total nitrogen inflow concentrations for the pre-summer period (winter, spring, summer) of each respective year. Further, we included the Birgean Work for spring and summer of each year as the Birgean Work represents the amount of external energy (mostly by wind shear stress) that is needed to build up the

current thermal structure. In addition to Birgean Work, we also included Schmidt Stability, the HBR ratio, the onset, end and duration of spring mixing, the onset, end and duration of summer stratification, the mean hypolimnetic water temperature at the onset of stratification, as well as the end and duration of the ice period prior to summer to investigate the effects of physical control on hypolimentic anoxia. In-lake biogeochemical processes were represented by the maximum height of anoxia during summer, the dissolved oxygen concentration differences between spring mixing onset and offset in the hypolimnion

(Livingstone and Imboden (1996) suggested that in eutrophic lakes dissolved oxygen reductions during the mixing phase can have profound effects on the summer anoxia), organic carbon (both dissolved and particulate, respectively) concentration differences between spring mixing and stratification in the hypolimnion, as well as cumulative gross primary production in the epilimnion and hypolimnion. Organic matter gradients were investigated because dissolved organic carbon can be used as a proxy for allochthonous organic matter contributions to bacterial mineralization rates (Hanson et al., 2003). Gross primary

production was included as an example organic matter source that can fuel bacterial mineralization (Yuan and Jones, 2019).

To determine the relative importance of these candidate predictors that may influence the duration and extent of anoxia, we applied the Boruta R package (Kursa and Rudnicki, 2010) to identify the relevant predictors by using a wrapper built around a random forest classifier. The Boruta feature selection duplicates predictor values, which are then randomly shuffled to create so-called shadow attributes. If the variable predictor values (here the averaged accuracy loss normalized by

the standard deviation, obtained from multiple random forest classifier runs) of the original values are significantly greater





than the shadow predictor values, these variables are deemed relevant (Kursa and Rudnicki, 2010). The first year, 1979, was dropped from the investigations due to a lack of prior winter information. Meteorological quarterly divisions (DJF, MAM, JJA, SON) of the year were used to define seasons. After selecting important predictors driving the inter-annual variability in Anoxic Factor using the random forest method, we applied the remaining 10 selected predictors in a multiple linear regression
model to quantify their respective importance on predicting the Anoxic Factor. Further, stepwise model-selection iteratively removed predictors to improve the regression model's AIC. Our final multiple linear regression model to predict Anoxic Factor included seven variables: HBR ratio during summer, maximum height of anoxia above sediment, Hypolimnetic water temperature at stratification onset, hypolimnetic GPP, Schmidt Stability in summer, Birgean Work in summer, and onset date of stratification (scaled predictors, adjusted $R^2 = 0.88$, p < 0.001).

$\hat{y} = -0.47 Summer.HBR - 0.15 Intensity + 0.29 Wtemp.Strat + 0.23 Epi.GPP + 0.53 Summer.St -$
$0.64 Summer.B - 0.66 Onset.Strat - 1.04 * 10^{-15} + \hat{\epsilon},$ (8)
where $\hat{\epsilon} \sim N(0, 34^2)$.

Relative importance of model fit was calculated as the $R^2$ contribution averaged over ordering among regressors (relaimpo package, Grömping 2006).

**3 Results**

**3.1 Oxygen Depletion Rates**

The derived annual oxygen depletion rates by the deductive model confirmed Lake Mendota's hypolimnetic anoxia as primarily driven by mineralization of organic matter. Observed oxygen depletion rates and area-volume ratios were positively correlated for all years except 1993, 1997 and 2007 (Figure 2). For years with a positive relationship, the average volume sink
$J_V$ as 0.16 g m$^{-3}$ d$^{-1}$ and the average area sink $J_A$ with 0.04 g m$^{-2}$ d$^{-1}$ (adjusted $R^2 = 0.13$, p < 0.001). Lake Mendota's hypolimnetic oxygen depletion is mainly driven by water column respiration processes over sediment oxygen demand. The annual volumetric depletions rate followed a normal distribution with an increase in the volumetric sink in recent years. The areal depletion rate distribution was positively skewed. An inspection of the residuals from the model fits indicates that the linear regression model may not be appropriate for some years, especially for values of the sediment to area volume ratio $\alpha(z)$ near
0.5 m$^2$ m$^{-3}$.

Averaging this total oxygen depletion rate (volume and area sink) over the hypolimnion, gave a potential total oxygen depletion of approx. 1 g m$^{-3}$ d$^{-1}$. To conceptualize such a depletion rate in our deterministic GLM-AED2 model, we used a maximum sediment oxygen demand (SOD) of 100 mmol [$O_2$] m$^{-2}$ d$^{-1}$, which represented the total sum of volumetric and areal oxygen sinks indirectly as we aimed to represent internal fluxes of organic carbon from the sediment back into the water
column that would additionally drive oxygen depletion. This high value of SOD was scaled by the water temperature using an Arrhenius multiplier, effectively reducing it to a value between 1 to 1.5 g m$^{-3}$ d$^{-1}$ of maximum oxygen depletion by the sediment





sink in the hypolimnion. A recent modeling study investigating the formation of metalimnetic oxygen minima in a drinking water reservoir by Mi et al. (2020) confirmed that such high maximum SOD values are typical for many lakes.

### 3.2 GLM-AED2 Calibration and Validation

The thermal characteristics of Lake Mendota were replicated well, especially water temperatures in the surface layers ( Table 2, Figure 3, Appendix Figure A2), with an RMSE of 1.3 °C, and an NSE and a KGE of 0.97, which is within the range of previous modeling studies (Bruce et al., 2018; Read et al., 2014). In contrast, the water quality model reproduced concentrations of the biogeochemical variables better at depth than at the surface, as evidenced by higher NSE values ( Table 2, Appendix Figure A3 – A7). The density distributions of residuals (observed minus simulated data) are in agreement

(Figure 4) for water temperature, dissolved oxygen concentrations, nitrate, phosphate and ammonia, whereas the model overestimated dissolved inorganic carbon concentrations and chlorophyll-a concentrations, and underestimated silica concentrations. It should be noted that the inlet concentrations of DIC and silica were assumed to be constant over the simulation period, probably causing the discrepancies between model results and observed data. An overview of the used and calibrated parameter values is given in Appendix Table A2.

### 330 3.3 Heat Budget Dynamics

Lake Mendota's annual stratification dynamics were characterized by a short spring mixing period followed by a very stable summer stratification period, which further promotes hypolimnetic oxygen depletion. On average, Schmidt Stability peaked in July at approx. 720 J m$^{-2}$ (Figure 5), followed by a peak in the Birgean Work at approx. 1250 J m$^{-2}$. A low Schmidt Stability value in spring close to zero was representative of the overturn period (period I, Figure 5). During this time, the Birgean Work,

as well as stored internal energy, increased rapidly, and the water column remained well oxygenated. The spring overturn period (period I) was characterized by low HBR values (ratio of St+B to B) with an average of 0.85 (MMA in Figure 6A). Low HBR denoted very unstable regimes due to an abundance of external energy compared to the required energy to keep the lake mixed. The start of the spring overturn period coincided with ice melt and open-water conditions, although in some years, the thermal structure of the lake was well mixed prior to ice off and spring overturn. May was the earliest month where the

average HBR was above 1 (Figure 6B), which indicated that the water layers below the thermocline became isolated from the surface layers. Following period I, Schmidt Stability increased in conjunction with the spatial extent of anoxia in the lake water column (Figure 5). The heat budgets, as well as the anoxic area, peaked during this second phase (period II) and declined, although the peak of anoxic area lagged behind the heat budget peaks. As the Schmidt Stability decreased to near zero in fall, mixing is initiated causing the water column to become oxygenated.

The stratification phase (period 2) had an average HBR value of 1.45, which indicated that an additional energy input of 45% would be needed to keep Lake Mendota isothermal during stratified summer conditions (Figure 6A). Lake Mendota's mean summer HBR value was similar to Lake Steinsfjord (max. depth 22 m, Kjensmo, (1994)) and was larger than the HBR values of the unstable lake systems Lake Marion (max. depth 4.5 m) and Lake Wingra (max. depth 6.1 m) (Kjensmo, 1994).



The oxygenation of the water column lagged behind the stratification period, and even when the Schmidt Stability values at
the end of the 2$^{nd}$ period were close to zero, a certain amount of the lake's area can remain anoxic.

Heat storage in Lake Mendota began after ice off and increased rapidly between the end of the mixing period and the
onset of stratification (Figure 7A). The amount of internal energy stored at the beginning of stratification correlated with the
maximum available amount that will be stored during the stratified period. Over the course of each year the amount of stored
internal energy was positively correlated with Schmidt Stability (Figure 7B). Nonetheless, the stored internal energy fluctuated
year-to-year. Birgean Work was also positively correlated with both Schmidt Stability and Internal Energy. The relationship
between Schmidt Stability and the spatial anoxia extent exhibited a clockwise hysteresis (Figure 7C). Beginning in June,
Schmidt Stability increased as stratified conditions established in the water column. Schmidt Stability peaked on average in
August. Simultaneously the anoxia height followed the progression of Schmidt Stability, but peaked in September. In Lake
Mendota, the anoxia height was limited by the thermocline depth, as the low vertical turbulent diffusivity of the thermocline
acted as a barrier for an encroachment of anoxic conditions into the surface mixed layer. Anoxia height decreased after
September with decreasing Schmidt Stability values. Thermocline depth and anoxia height declined in parallel until Schmidt
Stability reached zero. In Lake Mendota, as most lakes, the surface layer was the region of significant heat storage (Figure
7D). Once stratified, heat storage in deeper water layers was limited, whereas heat in the upper 5 m of the lake increased
throughout the summer, and accounted for up to 40% of the total internal energy stored during summer.

### 3.4 Oxygen Dynamics

Dissolved oxygen dynamics, including the spatial extent of oxygen depletion in the water column, and the timing of summer
anoxia periods, were replicated by the model (Figure 8A-B); although the model overestimated spring and summer time surface
oxygen concentrations due to a higher net ecosystem production. The depth-averaged fit criteria of dissolved oxygen
concentrations were similar to a recent study from Farrell et al. (2020) in which the RMSE in the epilimnion and hypolimnion
were about 1.88 mg/L and 2.49 mg/L, respectively. The model captured annual anoxia events in the hypolimnion (Figure 9A),
and the range of the simulated Anoxic Factor was similar to the derived Anoxic Factor from observed data (Figure 9B). The
model failed to replicate extreme events (for instance the very low Anoxic Factor in 2002) and did not capture a recent positive
trend of Anoxic Factor since 2010. The simulated Anoxic Factor averaged $56.7 \pm 5.2$ days with an RMSE of 7 days (correlation
coefficient r = 0.28).

### 3.5 Regression Model

We included in total 7 predictors in our linear regression which were deemed important by the Boruta algorithm and a stepwise
linear model investigation using AIC: Schmidt Stability and Birgean Work during summer, the intensity of anoxia (maximum
depth from the surface of anoxia in the water column), the onset date of stratification, gross primary production in the
epilimnion, the water temperature at stratification onset in the hypolimnion, as well as the HBR ratio in summer (Appendix
Table A3).





The linear model showed a good agreement between simulated and predicted Anoxic Factor (Figure 10 A, Appendix Table A3). The Anoxic Factor was positively correlated to the summer Schmidt Stability, the summer HBR ratio and the gross primary production in the epilimnion. It was negatively correlated to the summer Birgean Work, the water temperature at stratification onset, the maximum height of anoxia and the onset of stratification (Figure 10 B).

## 4 Discussion

### 4.1 Controls of Inter-annual variability on Hypolimnetic Anoxia

Inter-annual variability in the Anoxic Factor for Lake Mendota is influenced primarily by physical processes that regulate thermal and stratification dynamics, and less so by processes that influence organic matter. The timing of stratification, summer Schmidt Stability, and the summer ratio HBR all influence anoxic factor, and are all driven mainly by atmospheric drivers and 390 heat convection throughout the water column. The only significant predictor of anoxic factor directly related to biological processes is gross primary production in the epilimnion. Together, these variables explain 65% of the total relative variability in Anoxic Factor (Appendix Table A3). For eutrophic lakes, this suggests two critical points. First, climate has direct control over lake phenology. Climate drives the timing of stratification onset and stratification strength, and that controls the year-to-year variability in Anoxic Factor. Second, biology matters, but its interannual dynamics are not that influential, at least for this 395 eutrophic lake with a residence time greater than one year.

### 4.2 Physical Control over Anoxic Factor

Oxygen dynamics in Lake Mendota are strongly governed by the stratification strength in the water column. Snortheim et al. (2017) came to a similar conclusion, analyzing a shorter time period (2007-2010), arguing that changes in the atmospheric boundary conditions - air temperature, wind speed and relative humidity - are driving changes in the hypolimnetic anoxia 400 development of Lake Mendota. Here, we link these atmospheric drivers to changes in the water column's stratification (as quantified by Schmidt Stability and Birgean Work). Over our 37-year simulation, anoxia onset occurred in the days following stratification onset. During stratification, the establishment of a strong density gradient between the upper and the lower layers in the water column reduces vertical turbulent diffusivities and limits the downward flux of dissolved oxygen. Without any additional oxygen source (e.g., atmospheric fluxes or primary production), dissolved oxygen concentrations below the 405 thermocline are rapidly consumed by bacterial mineralization of organic matter in the water column and sediment.

In Lake Mendota, the temporal and spatial extent of anoxia is limited by the length of the summer stratification period (e.g. onset and offset of stratification, heat storage in water column prior to stratification, see Figure 5), the stratification strength and thermocline depth (e.g. Schmidt Stability, wind shear stress, see Figure 7), respectively. The number of days between the onset of spring mixing, which begins immediately following ice off, and summer stratification, determines the 410 maximum amount of internal energy stored in summer. An early spring overturn and a slightly later stratification start would lead to increased anoxia height in the water column, though not necessarily a higher Anoxic Factor as the duration of anoxia





could be shorter. The mixing period is essentially a turning point for the gradient of the internal heat accumulation, which increases rapidly following mixing. Still, as most energy is stored in a thin surface layer, short-duration extreme wind events or cold weather periods can deplete that additional stored heat before summer stratified conditions are reached. The storage of

heat simultaneously increases Birgean Work, and later Schmidt Stability, increasing the resistance of the water column to mixing and limiting vertical fluxes from the epilimnion to the hypolimnion and vice versa. In summer, a higher amount of stored internal energy is also related to a higher Schmidt Stability, further increasing the spatial extent of anoxia. Ultimately, the spatial extent of anoxia is limited by the thermocline depth, as in all simulated years the anoxia height reaches a maximum during late summer when the thermocline depth was already deepening.

## 420 4.3 Biological Control over Anoxic Factor

Gross primary production (GPP) in the epilimnion prior to summer stratification is a secondary, but still important predictor of anoxia. GPP fuels the sinking of particulate organic carbon (POC) into deeper layers before the establishment of a thermocline. In the hypolimnion, POC is readily decomposed into DOC and mineralized by bacteria in the numerical model, and reflects the dissolved oxygen volume sink. Unexpectedly, factors controlling year-to-year variation in GPP, such as

external loadings of nutrients (specifically nitrogen and phosphorus), are not evident in the anoxia patterns in Lake Mendota. This is likely due to the historically high autochthony of the eutrophic lake, with phytoplankton blooms document back to the early 1900s (Lathrop, 2007), and importance of autochthony over allochthony (Hart, 2017), thereby minimizing the need for external nutrient loads to stimulate phytoplankton production. While biological contributions to volumetric and sediment oxygen demands are well-described for a broad range of lakes (Gelda and Auer, 1996; Matzinger et al., 2010; Müller et al.,

2012; Rippey and McSorley, 2009; Yuan and Jones, 2019), for eutrophic lakes, the control over available organic substrate for hypolimnetic oxygen demand may depend more on internal processing (autochthony) than external subsidies (allochthony).

The regression models showed that variables related to load dynamics were not significant predictors of Anoxic Factor over nearly four decades. The total phosphorus and nitrogen loads, the change in dissolved oxygen during spring overturn, the temporal change in organic carbon pools, and ice duration were not found important based on the random-forest classifier.

Phosphorus cycling in Lake Mendota is complex, so it may not be surprising that load dynamics in any one year are, to a certain extent, uncoupled from the hypolimnetic oxygen demand (Hanson et al., 2020). The relatively long water residence time of Lake Mendota (approx. 4 years, McDonald and Lathrop, 2017), along with the high internal phosphorus loading rate, means that external phosphorus loads represent only about 1/3 of the available phosphorus in the epilimnion (Soranno et al., 1997). Furthermore, high primary production rates that exceed the total lake mineralization, along with external loads of

organic carbon, lead to a high storage of organic matter in the sediments that can likely carry over from one year to the next (Hart, 2017). In a more nutrient-poor system, the nutrient and carbon availability would likely be more important predictors.

The model replicated the extreme maximum anoxia event in 1998 but struggled to replicate the minimum in 2002. The discrepancies of 5-10 days between the simulated and observed range of the Anoxic Factor beginning in 2010 are related to an increased spatial as well as temporal extent of summer anoxia (Appendix Figure A8), which was not captured by the





model. The change in Anoxic Factor post-2010 may be due to an ecosystem shift in Lake Mendota that began in 2009, when
the invasive spiny water flea (*Bythothrephes longimanus*) was detected in surprisingly high densities in the lake (Walsh et al.,
2016b, 2018). Spiny water flea effectively became the dominant daphnia grazer, causing historically low daphnia biomass in
2010, 2014 and 2015 (Walsh et al., 2016a) and reducing water clarity. The impact of spiny water flea on specific phytoplankton
groups may have increased organic matter supply to the hypolimnion by grazing. Our GLM-AED2 model could not replicate
this food web change, and subsequent shift in anoxia dynamics, due to limitations of the numerical model, i.e., GLM-AED2
had constant ecological parameters over the entire modeling period and did not have zooplankton dynamics instantiated.

The simple deductive model established that the volumetric oxygen sink (i.e. water column oxygen demand) is
consistently higher (on average about four times higher) than the sediment oxygen sink. The volumetric sink in lakes has been
found to be strongly dependent on the trophic state of the lake, whereas the sediment sink is not (Rippey and McSorley, 2009).
Eutrophic lakes tend to have high volume sinks that reach maxima of about 0.23 g m$^{-3}$ d$^{-1}$ (Rippey and McSorley, 2009) similar
to the average volume sink of 0.16 g m$^{-3}$ d$^{-1}$ quantified by the deductive model for Lake Mendota. This finding is confirmed
by the works of Conway (1972) who found that the high hypolimnetic oxygen demand of Lake Mendota was driven by algae
decomposition, originating from the surface layer. Although eutrophic lakes tend to have a high sediment oxygen demand, the
specific values can range from 0.3 g m$^{-2}$ d$^{-1}$ (Romero et al., 2004; Steinsberger et al., 2019) to extreme values of 80 g m$^{-2}$ d$^{-1}$
(Cross and Summerfelt, 1987), most studies measured or applied a value between 1 to 4 g m$^{-2}$ d$^{-1}$ (Mi et al., 2020; Veenstra
and Nolen, 1991). The sediment oxygen demand calculated by our deductive model of 0.04 g m$^{-2}$ d$^{-1}$ was closer to the average
value of approx. 0.08 g m$^{-2}$ d$^{-1}$ measured by Rippey and McSorley (2009) on 32 lakes.

### 4.4 Improving the Modeling Framework

The coupled GLM-AED2 model was able to generally replicate the thermal dynamics and biogeochemistry of Lake Mendota.
In contrast to the calibration of Lake Mendota by Bruce et al. (2018) using an earlier version of GLM (v. 2.2.0), our model
reproduced the water temperatures in the surface layer better than the bottom layer dynamics (RMSE for epilimnion and
hypolimnion water temperatures, respectively, from Bruce et al., 2018: 1.94 °C and 1.42 °C). This is probably due to the close
proximity of the atmospheric forcing boundary condition to the surface layers, whereas the energy balance approach used by
GLM potentially underestimates vertical mixing, and hence overpredicts bottom layer water temperatures. In contrast, the
model achieved better fits of the biogeochemical variables in the bottom layer. Better fits in the hypolimnion were likely
achieved through directed calibration of sediment fluxes during the calibration-validation approach. The implementation and
testing of alternative vertical mixing schemes for the Lake Mendota model (e.g. vertical mixing using a k- ε turbulence model)
could potentially improve vertical transport and water temperature dynamics in deep layers. Further, using transient sediment
boundary conditions with dynamic parameters over time could improve the model fit with the observed data, and could
replicate potential ecosystem shifts. As the spatial extent of hypolimnetic anoxia is fundamentally three-dimensional (Biddanda
et al., 2018), fully resolving anoxia in space and time likely requires a 3D model (Bocaniov and Scavia, 2016). Still, such a
model has higher computational needs for long-term calibration-validation analysis, and current monitoring is inadequate to





validate the results as most measurements are only made at the deepest point of the lake. Therefore, additional monitoring sites would need to be established. Improved spatial monitoring would be useful in validating our 1D approach and setting up higher
dimensional numerical models.

Our calibrated GLM-AED2 model overestimated spring phytoplankton biomass, which resulted in overestimation of surface dissolved oxygen concentrations. This primary overproduction is a potential source of uncertainty for the anoxia timing below the thermocline as the model's anoxia dynamics lag behind the observed ones. The time difference between the simulated and observed dissolved oxygen decline below the thermocline during stratification could be explained by an
underestimation of sinking simulated organic material into the hypolimnion. Discrepancies between simulated and observed Anoxic Factors, therefore, could be rooted in our simplifications of the phytoplankton dynamics and the related organic matter fluxes, and highlight the importance of improving the representation of phytoplankton and zooplankton dynamics in numerical models. Simulating a magnitude of individual species rather than functional phytoplankton groups has been shown to improve numerical water quality and ecosystem predictions (Hellweger, 2017), though it is unclear if it could improve spring bloom
predictions in Lake Mendota. Further, better numerical representations of phytoplankton life cycles (Hense, 2010; Shimoda and Arhonditsis, 2016), and/or allometric scaling (Shimoda et al., 2016) could significantly improve numerical phytoplankton predictions. It is noteworthy that biweekly monitored data of Lake Mendota required interpolation of the observed data in order to calculate the observed Anoxic Factors. This adds uncertainty to the observed Anoxic Factor, as monitoring likely missed important daily (or even sub-biweekly) fluctuations in dissolved oxygen.

It should be noted that as a statistical approach, the deductive regression model does not account for important mechanisms that may explain nonlinearities in the hypothetical linear relationships between oxygen depletion rate and the sediment to volume rate. Thus, the deductive regression model may be biased for Lake Mendota. As the model still advanced our broader system understanding by quantifying the range of the sediment oxygen demand, it was still helpful to investigate observed dissolved oxygen concentration data.

**4.5 Implications for landscape and climate change**

The strong relationship between anoxia and water column stability suggests that a changing climate might increase Anoxic Factor. Future climate in the region is expected to warm (Veloz et al., 2012), which may amplify and prolong water column stratification through increasing air temperatures (O'Reilly et al., 2015; Winslow et al., 2017). Shorter ice duration or even the total loss of ice (Sharma et al., 2019) could promote earlier heat storage in Lake Mendota, which could potentially increase
summer Schmidt Stability, as demonstrated by (Farrell et al., 2020)). Earlier heat accumulation would cause a stability increase and an earlier onset of stratification, thereby extending the duration of anoxia. Earlier onset of anoxia may cause the anoxia height to be spatially limited by an earlier, and therefore lower, thermocline depth. Therefore, warming air temperatures will likely increase the Anoxic Factor of Lake Mendota through prolonged temporal and increased spatial extent of anoxia. It is worth noting that the correlation between Anoxic Factor and the water temperature in the hypolimnion at stratification onset
was weakly negative. Higher water temperatures in a mixed water column prior to stratification onset are related to less stable





stratified summer conditions. This feedback, potentially enhanced by shorter ice periods and warmer spring overturn periods, could shorten the extent of summer anoxia.

Although our model evaluation supported the claim that external phosphorus loads are not important predictors of inter-annual variability in anoxia, future changes in the landscape (Motew et al., 2019), e.g. reduced agricultural application

of phosphorus, less direct run-off pathways from the catchment to the lake, or more urbanization, may change these relationships. Lakes with nutrient concentrations lower than Lake Mendota would almost certainly experience higher primary production with elevated nutrient loads, and higher primary production would likely fuel higher hypolimnetic respiration (Rippey and McSorley, 2009). Thus, the link between catchment processes and lake anoxia, which was not detectable in this study, would likely be important in lakes with meso- or oligotrophic states (Farrell et al., 2020). For Lake Mendota, the only

reasonable management approach to reducing anoxia is to lower nutrient loads, especially given that anoxia duration in Lake Mendota is related to thermal stratification which is predicted to increase with future warmer air temperatures.

**5 Conclusions**

We presented a novel modeling framework combining three complementary approaches (deductive model, numerical GLM-AED2 model, and regression model) to conceptually identify the important drivers of year-to-year variability in the spatial and

temporal summer hypolimnetic anoxia extent of eutrophic Lake Mendota over a period spanning nearly four decades. Physical metrics – onset date of stratification, summer Schmidt Stability, and the ratio of St+B to B – were the most important predictors driving the summer Anoxic Factor. Although the gross primary production was still influential in affecting year-to-year variability of hypolimnetic anoxia, biological control over the Anoxic Factor was limited in our study period. As climate change is positively correlated to lake stratification characteristics (earlier, longer and more intense summer stratification), we

expect an increase in the Anoxic Factor of Lake Mendota in coming decades. The only local management option to mitigate future hypolimnetic anoxia in Lake Mendota is a reduction of external nutrient loads, which aims at shifting the lake towards oligotrophic conditions. As managers and decision makers undertake forward planning to guard against a decline in lake water quality as a result of climate change, decision support tools that support an understanding of lake dynamics over the long term are essential. The modeling framework developed here can be extended by an advanced sediment diagenesis model and an

uncertainty analysis, e.g. Bayesian analysis, to develop greater insight into effective strategies to mitigate environmental degradation.

**Code and data availability**

The data to run the 37-year simulation, including configuration files, driver data and model source code, as well as the simulation output used in this study are available at the Environmental Data Initiative (https://portal-

s.edirepository.org/nis/mapbrowse?scope=knb-lter-ntl&identifier=389&revision=1, Ladwig et al., 2020).



## Author Contributions

PCH, HAD, CCC, KMC and RL designed the conceptual model study and aim. HAD pre-processed the input data. RL and PCH performed the lake model simulations. YZ, LS and CD performed the catchment model simulations. RL analysed the data and prepared the manuscript with contributions from all co-authors.

## 545  Acknowledgments

The project was funded through an NSF ABI development grant (#DBI 1759865), an NSF CNH grant (#1517823), as well as NSF grants DEB 1753639 and DEB 1753657. Lake Mendota data were obtained from the North Temperate Lakes Long Term Ecological Research program (#DEB-1440297). We are thankful for supplementary dissolved oxygen field data from 1992-1994 by Patricia Soranno.

## 550  Competing Interests

The authors declare that they have no conflict of interest.

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





**Figures**

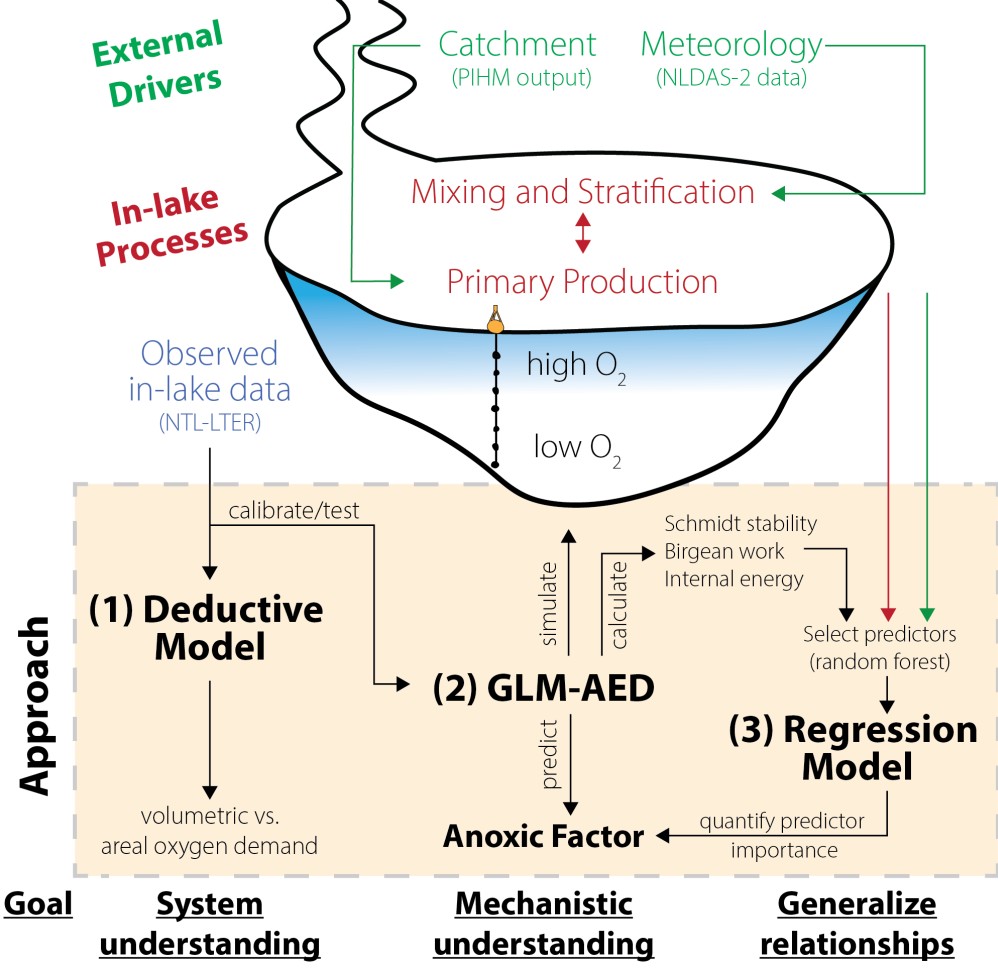

**Figure 1 Schematic overview of the modeling workflow: application of a (1) deductive model to further our system understanding about oxygen sink terms; replication of Lake Mendota using (2) GLM-AED2 to investigate hydrodynamic and ecosystem mechanisms; and application of (3) regression models to quantify the importance of ecosystem predictors on the Anoxic Factor.**






**Figure 2** Regression plots of the morphometric function α(z) against oxygen depletion rates for the years 1992 to 2018, which were calculated from temporally linearly interpolated observed data. The respective equations represent weighted linear regressions.

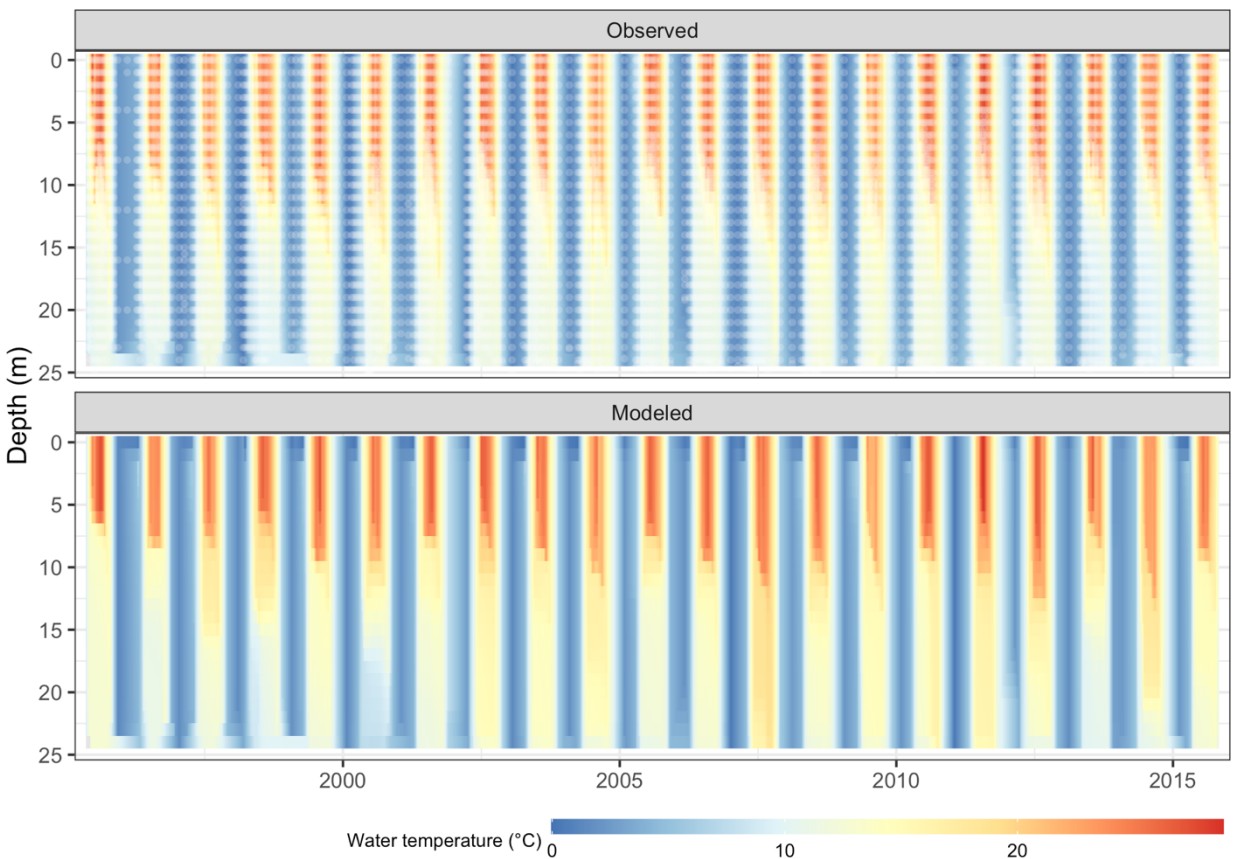

**Figure 3 Water temperature from observation (upper figure, white dots mark sample events) and GLM-AED2 simulation**





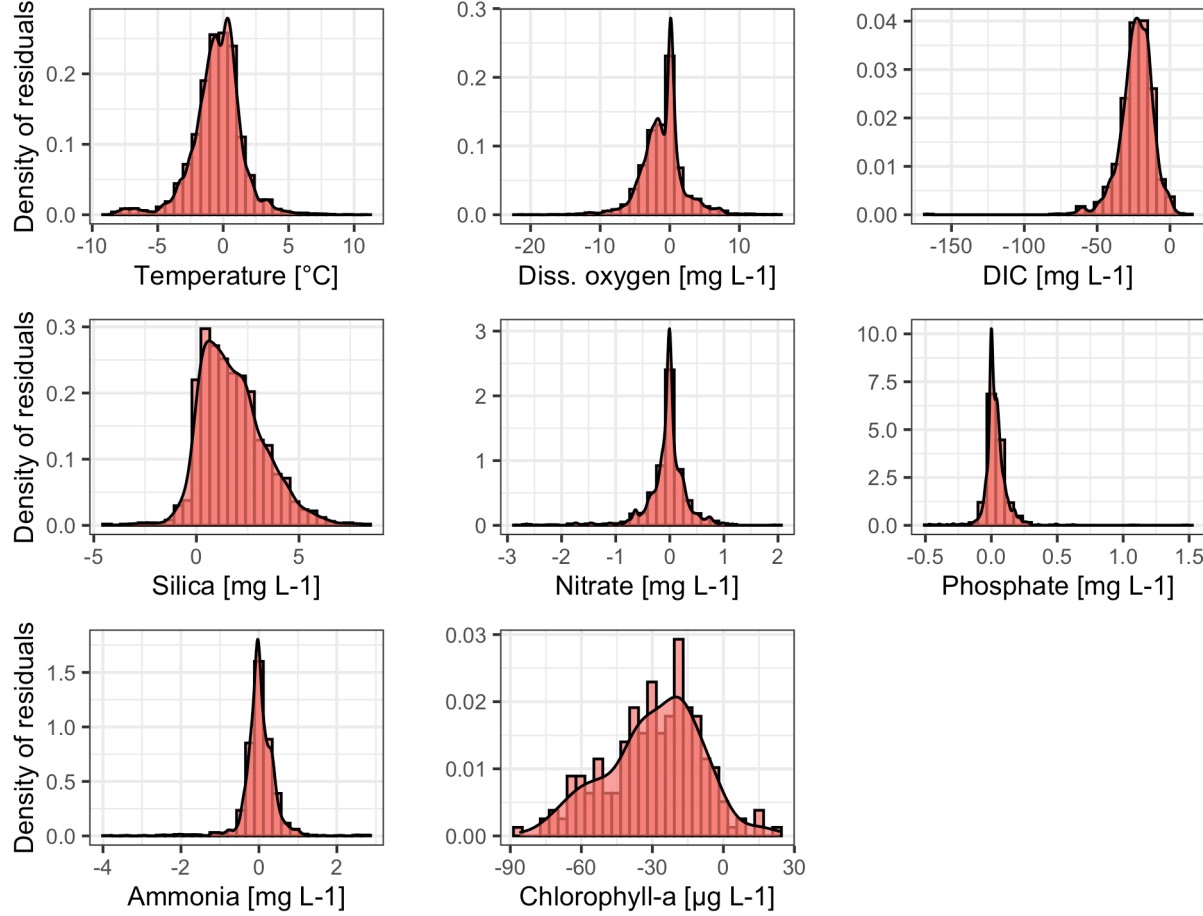

**Figure 4 Density distributions of residuals (observed - modeled) for water temperature, dissolved oxygen, dissolved inorganic carbon (DIC), silica, nitrate, phosphate, ammonia and phytoplankton. The density distributions include residuals over all data points (over each time-step over each depth), calculated from observations minus model predictions.**






**Figure 5 Daily average values of Schmidt Stability, Birgean Work, internal energy and anoxic area (below 1 mg L⁻¹) plus/minus the respective standard deviations (dashed lines) (internal energy is given in 10e6 J m⁻²). [Anoxic area units were adjusted for display purposes.]**





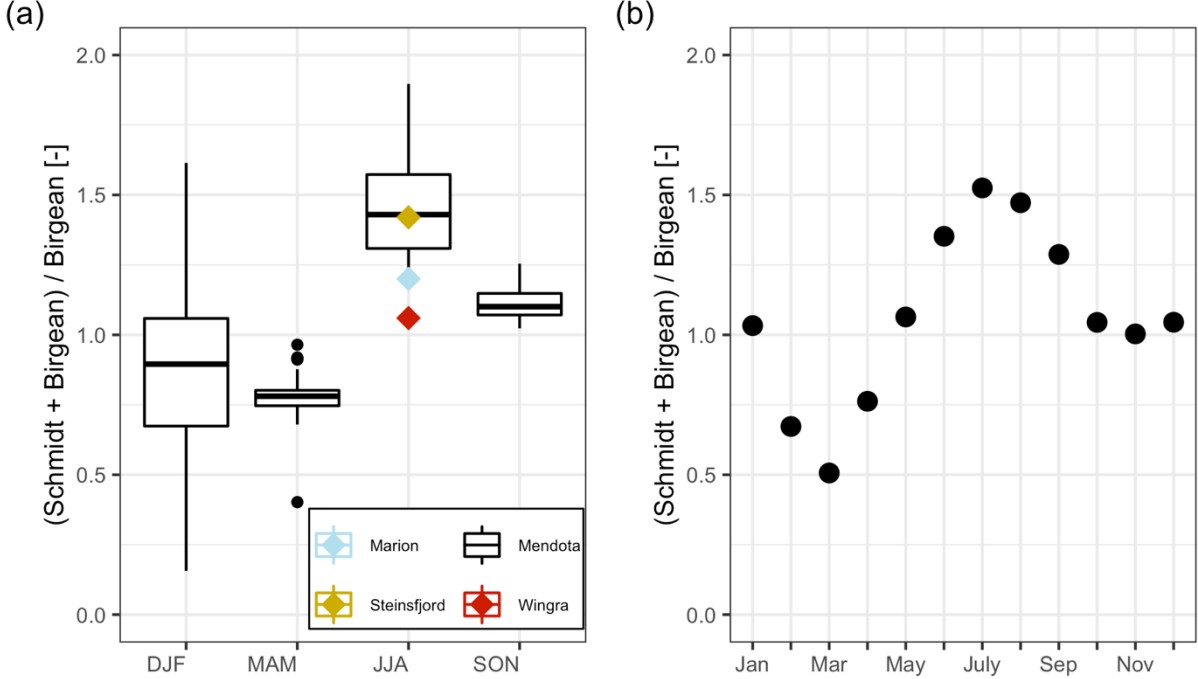


**Figure 6 Dynamics of average HBR = (St+B)/B over time. A box-plots of HBR over the meteorological seasons, which represent seasonal quarters of the year, beginning in December. The summer HBR values for Lake Marion, Lake Steinsfjord and Lake Wingra were taken from Kjensmo (1994). B Scatter plot of average HBR values for each month**





(a)

(b)

(c)

(d)





**Figure 7 Stored heat dynamics and relationships to stratification strength, thermocline depth and anoxia height. A: Time series of internal energy at the respective dates of ice off, mixing onset, mixing offset, stratification onset and stratification offset. B: Scatter plot of internal energy against Schmidt Stability whereas the color represents the magnitude of Birgean Work. C: Scatter plot of anoxia height against Schmidt Stability. The black line represents the average dynamic over the course of a year with the respective months as labels. The color corresponds to the thermocline depth in meters above the sediment. D: Time series of daily averaged internal energies stored over different depths and Schmidt Stability). The main heat storage happens in a shallow surface layer effectively after ice off and the simultaneous onset of a mixing period.**

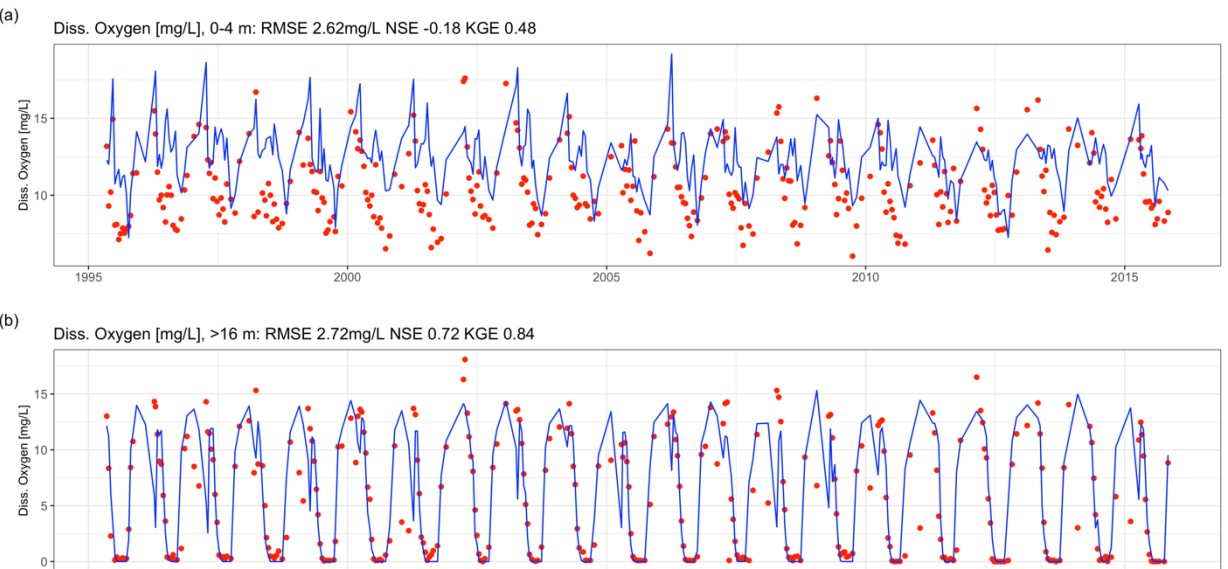

**Figure 8 Time-series comparison between observed (red dots) and modeled dissolved oxygen concentrations (blue lines). The fit criteria root-mean square error (RMSE), Nash-Sutcliffe coefficient of efficiency (NSE) and Kling-Gupta coefficient of efficiency (KGE). A averaged dissolved oxygen concentrations in the epilimnion (0-4 m). B averaged dissolved oxygen concentrations in the hypolimnion (deeper than 16 m).**





(a)



(b)

**Figure 9 Comparison of observed to modeled dissolved oxygen concentrations and ecosystem response. A Contour plot of observed (upper figure, white dots mark sample events) and simulated dissolved oxygen concentrations. B Comparison of simulated Anoxic Factor (red dots) against interpolated range of Anoxic Factor derived from observed data (box-whisker plots)**



(a)

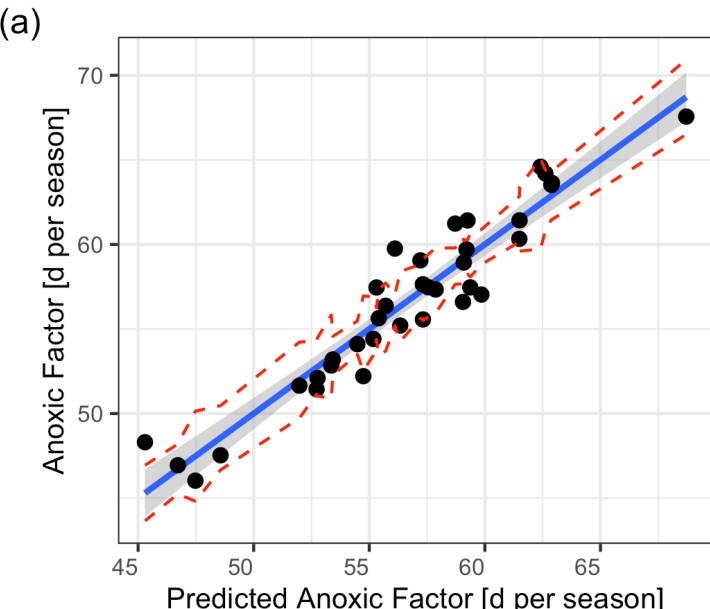

(b)

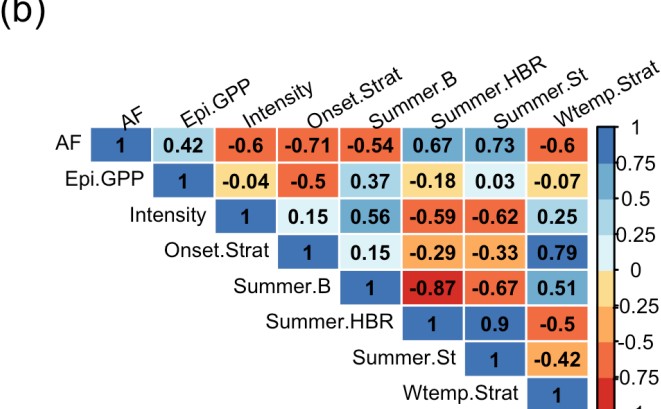

**Figure 10 Predicted against simulated summer Anoxic Factor. A Linear model with a prediction which was done using a multiple linear regression model of the form:** $\hat{y} = -0.47 Summer.HBR - 0.15 Intensity + 0.29 Wtemp.Strat + 0.23 Epi.GPP + 0.53 Summer.St - 0.64 Summer.B - 0.66 Onset.Strat - 1.04 * 10^{-15} + \hat{\epsilon}$, **where** $\hat{\epsilon}\,N(0, 34^2)$ **. The red lines represent confidence intervals. B Correlogram of the input data using Pearson correlation coefficients**

**Tables**

**Table 1 Overview of investigated predictors in a linear regression model on estimating the Anoxic Factor**



| Candidate predictor | Temporal period | Method | Unit |
|---|---|---|---|
| Schmidt Stability | Summer of year n | See section 'Post-processing of model output' | J per $m^2$ |
| Schmidt Stability | Spring of year n | See section 'Post-processing of model output' | J per $m^2$ |
| Birgean Work | Summer of year n | See section 'Post-processing of model output' | J per $m^2$ |
| Birgean Work | Spring of year n | See section 'Post-processing of model output' | J per $m^2$ |
| Ratio HBR | Summer of year n | See section 'Post-processing of model output' | - |
| Ratio HBR | Spring of year n | See section 'Post-processing of model output' | - |
| Onset, end and duration of spring mixing | Spring/summer of year n | Determined by Schmidt Stability values (close to or at zero, indicating mixed conditions) | Day of the year, or days |
| Onset, end and duration of summer stratification | Spring/summer/fall of year n | Stratification was defined when density difference between surface and bottom layer was above 0.1 kg $m^{-3}$ and surface temperature was above 4 °C | Day of the year, or days |
| Maximum height above sediment of anoxia | Summer of year n | Extracted from simulation output | m above sediment |
| End and duration of ice period | Winter/spring of year n-1 and n | Extracted from simulation output | Day of the year, or days |
| Dissolved oxygen difference between mixing end and mixing onset in the lower layer | Spring/summer of year n | Extracted from simulation output | mmol $O_2$ per $m^2$ d |
| Dissolved organic carbon gradient between stratification onset and mixing onset in the lower layer | Spring/summer of year n | Extracted from simulation output | mmol C per $m^2$ d |
| Particulate organic carbon gradient between stratification onset and mixing onset in the lower layer | Spring/summer of year n | Extracted from simulation output | mmol C per $m^2$ d |



| Mean water temperature in the lower layer at the onset of stratification | Summer of year n | Extracted from simulation output | °C |
| Total phosphorus inflow concentration | Winter/spring/summer of year n-1 and n | Extracted from driver data | g P per m$^2$ |
| Total nitrogen inflow concentration | Winter/spring/summer of year n-1 and n | Extracted from driver data | g N per m$^2$ |
| Cumulative gross primary production in the upper layer | Winter/spring of year n-1 and n | Extracted from simulation output | mmol per m$^2$ per day |
| Cumulative gross primary production in the lower layer | Winter/spring of year n-1 and n | Extracted from simulation output | mmol per m$^2$ per day |


**Table 2 Model performance for water temperature, dissolved oxygen, dissolved inorganic carbon, silica, nitrate, ammonia and phosphate. During calibration and validation, only the total fits over all depths and time-steps were calculated. Surface layers refers to a depth of 0 m below water table, and bottom layer to a depth of 20 m below water table. Fits for surface and bottom layer during calibration and validation are not shown as the fit over the whole water column and over time were used.**

| Variable | Calibration (2005 – 2015) | | | Validation (1995 – 2004) | | | Total period (1995 – 2015) | | |
|---|---|---|---|---|---|---|---|---|---|
| | RMSE | NSE | KGE | RMSE | NSE | KGE | RMSE | NSE | KGE |
| Water temperature [°C] | 2.26 | 0.87 | 0.92 | 1.78 | 0.92 | 0.96 | 1.96 | 0.91 | 0.94 |
| surface layer | | | | | | | 1.30 | 0.97 | 0.97 |
| bottom layer | | | | | | | 2.43 | 0.20 | 0.71 |
| Dissolved oxygen [mg L$^{-1}$] | 3.33 | 0.54 | 0.76 | 3.29 | 0.53 | 0.76 | 3.22 | 0.56 | 0.77 |
| surface layer | | | | | | | 2.77 | -0.36 | 0.46 |
| bottom layer | | | | | | | 3.31 | 0.64 | 0.81 |
| Dissolved inorganic carbon [mg L$^{-1}$] | 25.87 | -7.03 | 0.18 | 15.41 | -8.65 | 0.20 | 25.92 | -10.13 | 0.20 |
| surface layer | | | | | | | 19.71 | -10.79 | 0.22 |
| bottom layer | | | | | | | 29.19 | -12.84 | 0.15 |
| Silica [mg L$^{-1}$] | 2.83 | -1.32 | -4.55 | 1.42 | -0.55 | -1.77 | 2.33 | -0.83 | -3.10 |
| surface layer | | | | | | | 1.61 | -1.32 | -6.53 |
| bottom layer | | | | | | | 2.78 | -0.97 | -0.90 |
| Nitrate [mg L$^{-1}$] | 0.36 | -0.01 | 0.56 | 0.45 | -1.44 | 0.26 | 0.40 | -0.40 | 0.44 |
| surface layer | | | | | | | 0.35 | -0.18 | 0.49 |
| bottom layer | | | | | | | 0.30 | 0.29 | 0.34 |
| Ammonia [mg L$^{-1}$] | 0.60 | -3.03 | 0.17 | 0.48 | -3.28 | 0.15 | 0.56 | -3.05 | 0.17 |
| surface layer | | | | | | | 0.25 | -1.76 | 0.08 |
| bottom layer | | | | | | | 0.64 | 0.41 | 0.70 |





| | | | | | | | | | |
|---|---|---|---|---|---|---|---|---|---|
| Phosphate [mg L$^{-1}$] | 0.10 | 0.56 | 0.51 | 0.09 | 0.59 | 0.62 | 0.09 | 0.56 | 0.51s |
| surface layer | | | | | | | 0.03 | 0.40 | 0.43 |
| bottom layer | | | | | | | 0.17 | 0.40 | 0.36 |


## Appendix A

*Correspondence to*: Robert Ladwig (rladwig2@wisc.edu)

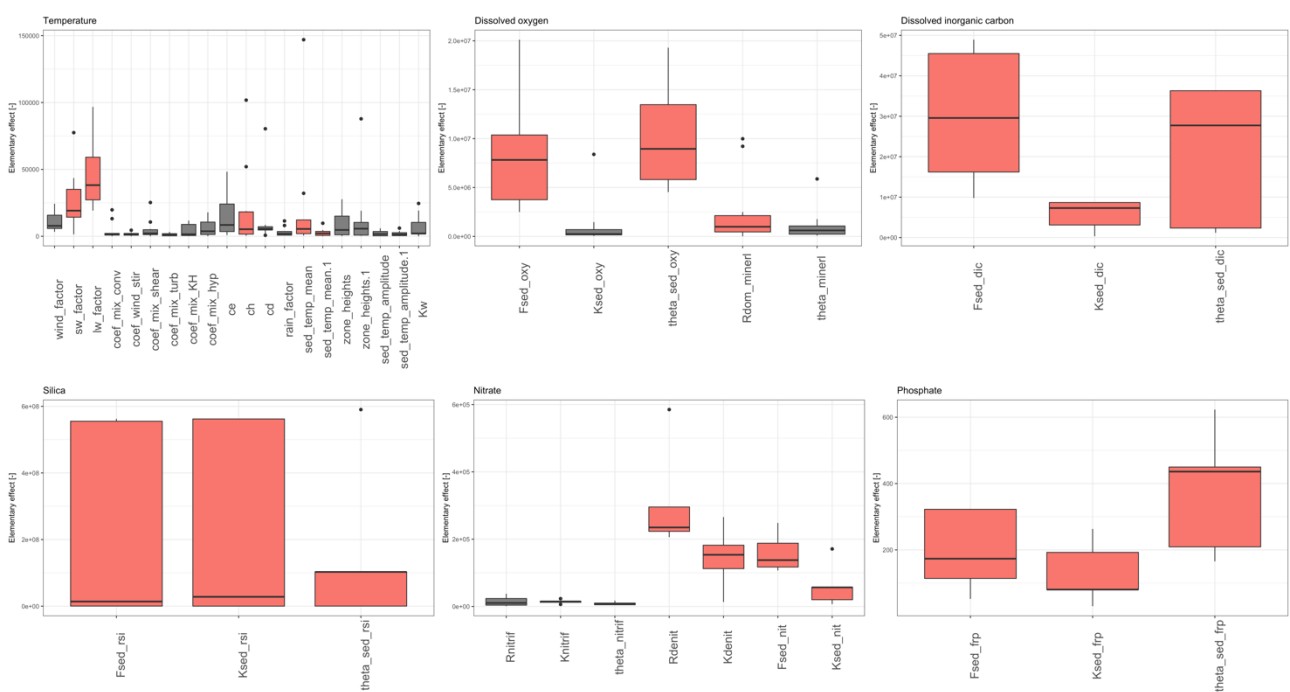

**Figure A1 Mean and standard deviations of absolute elementary effects quantified by the Morris Method for water temperature,**
**dissolved oxygen, dissolved inorganic carbon, silica, nitrate and phosphate. Colored bars are sensitive parameters that were used in**
**the calibration.**





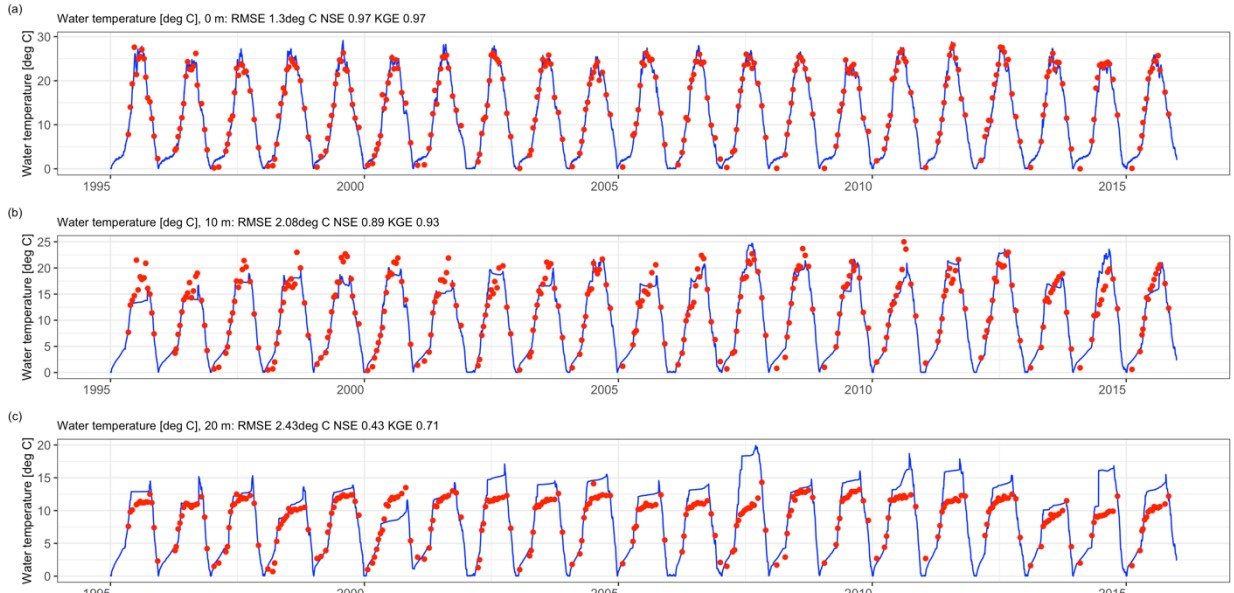

**Figure A2 Time-series comparison between observed (red dots) and water temperatures (blue lines). The fit criteria root-mean square error (RMSE), Nash-Sutcliffe coefficient of efficiency (NSE) and Kling-Gupta coefficient of efficien**

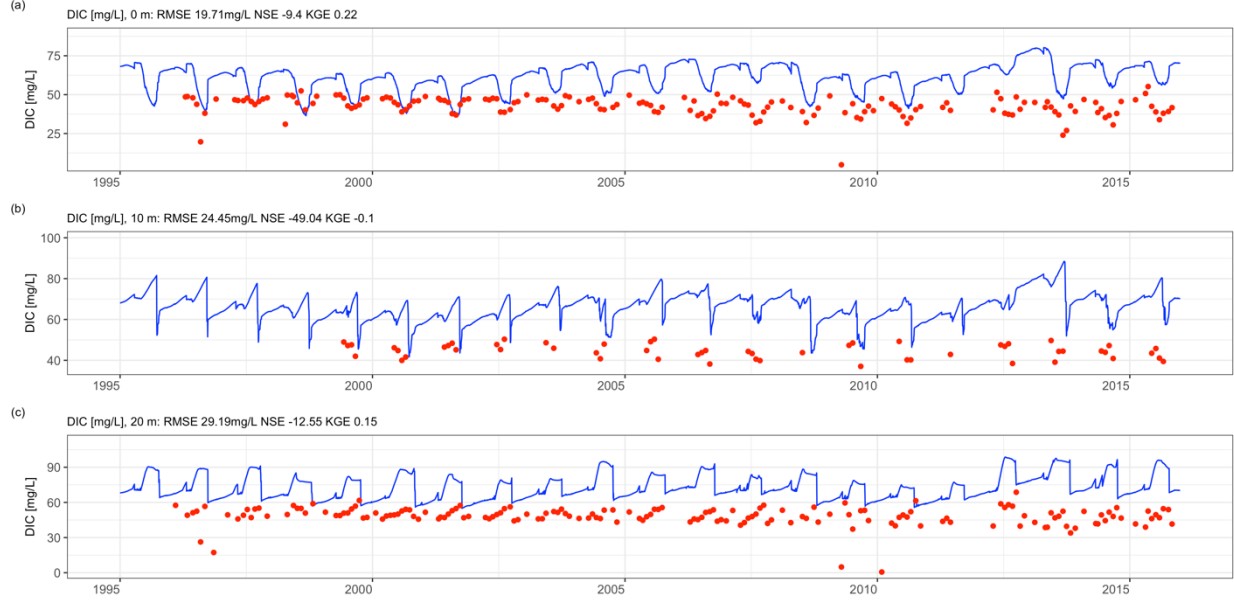


**Figure A3 Time-series comparison between observed (red dots) and modeled dissolved inorganic carbon concentrations (blue lines). The fit criteria root-mean square error (RMSE), Nash-Sutcliffe coefficient of efficiency (NSE) and Kling-Gupta coefficient of efficiency (KGE).**



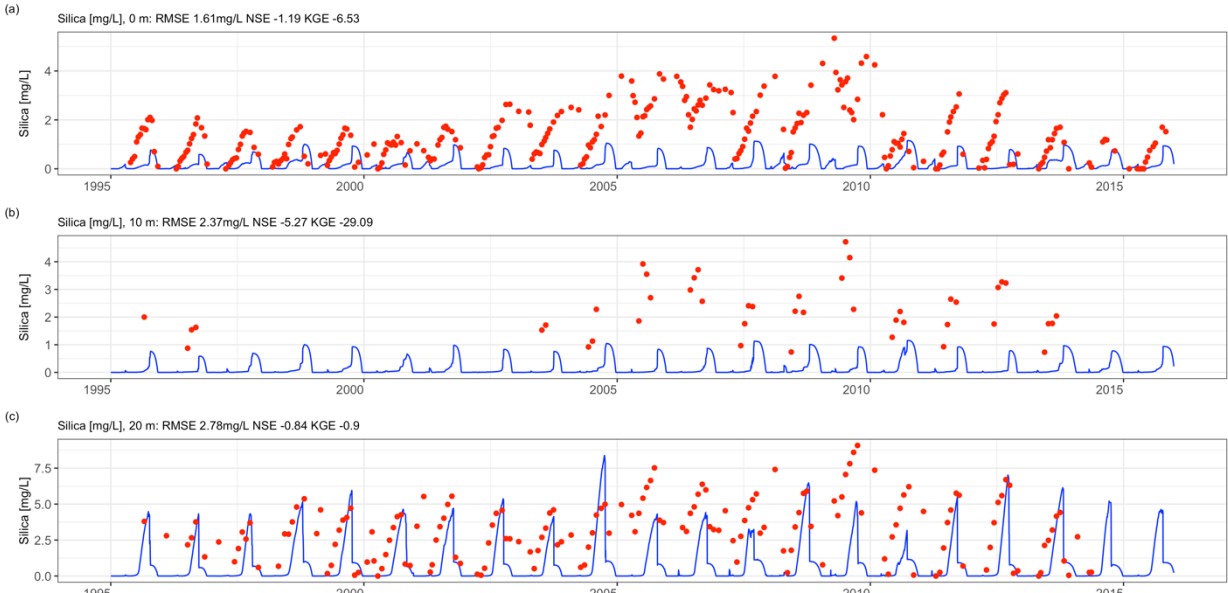

**Figure A4 Time-series comparison between observed (red dots) and modeled silica concentrations (blue lines). The fit criteria root-mean square error (RMSE), Nash-Sutcliffe coefficient of efficiency (NSE) and Kling-Gupta coefficient of efficiency (KGE).**

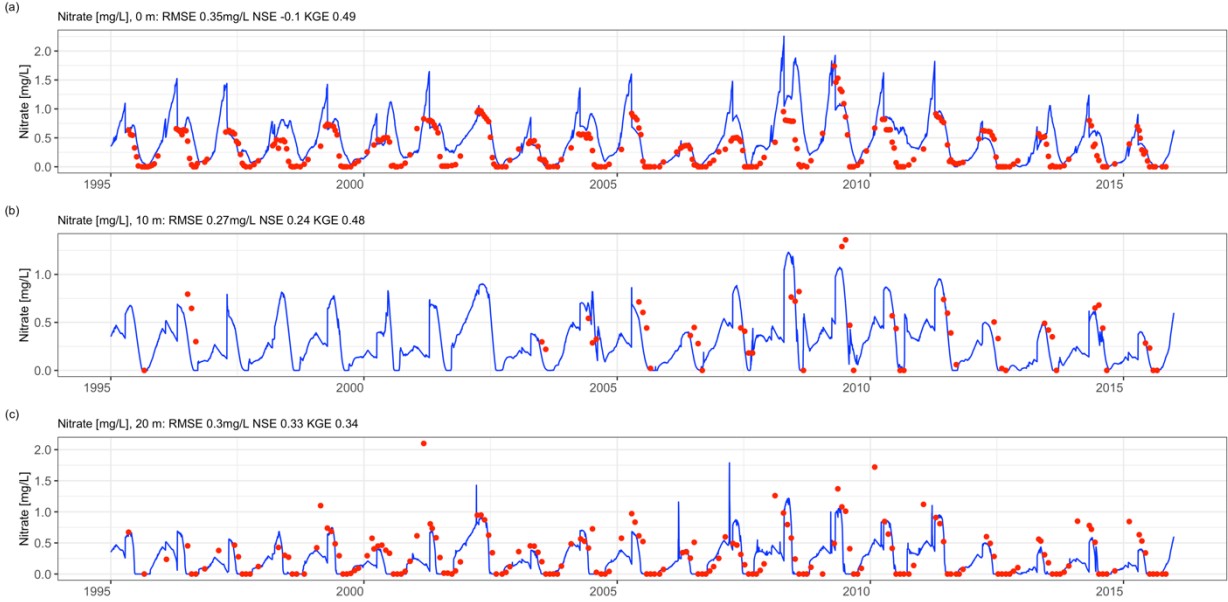

**Figure A5 Time-series comparison between observed (red dots) and modeled nitrate concentrations (blue lines). The fit criteria root-mean square error (RMSE), Nash-Sutcliffe coefficient of efficiency (NSE) and Kling-Gupta coefficient of efficiency (KGE).**





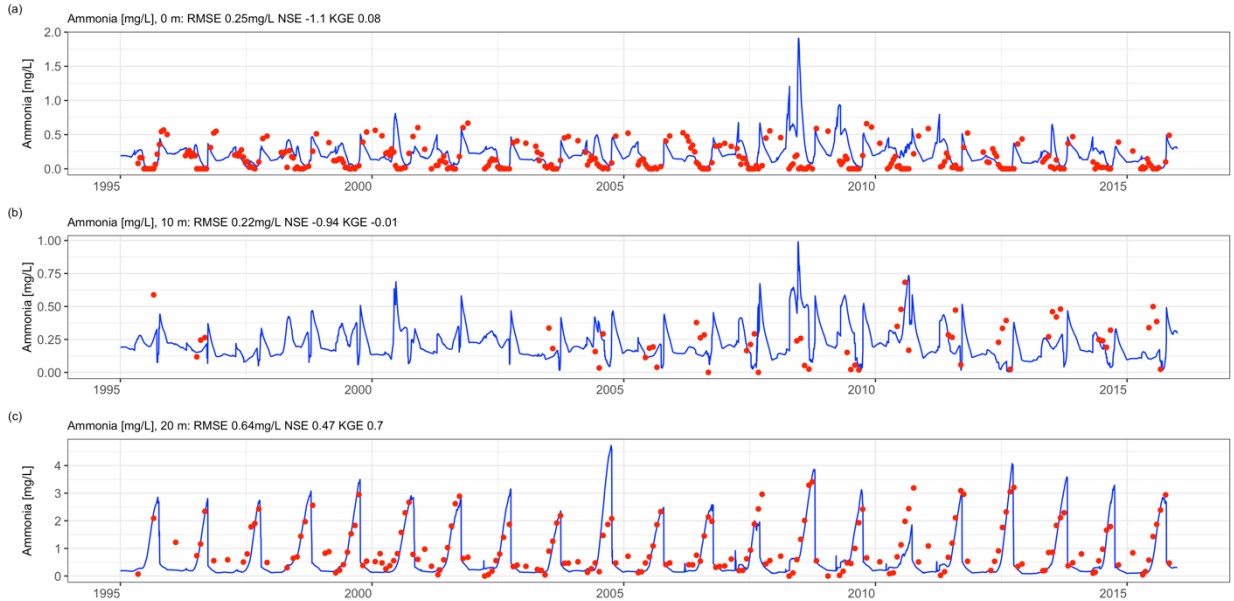


**Figure A6** Time-series comparison between observed (red dots) and modeled ammonia concentrations (blue lines). The fit criteria root-mean square error (RMSE), Nash-Sutcliffe coefficient of efficiency (NSE) and Kling-Gupta coefficient of efficiency (KGE).

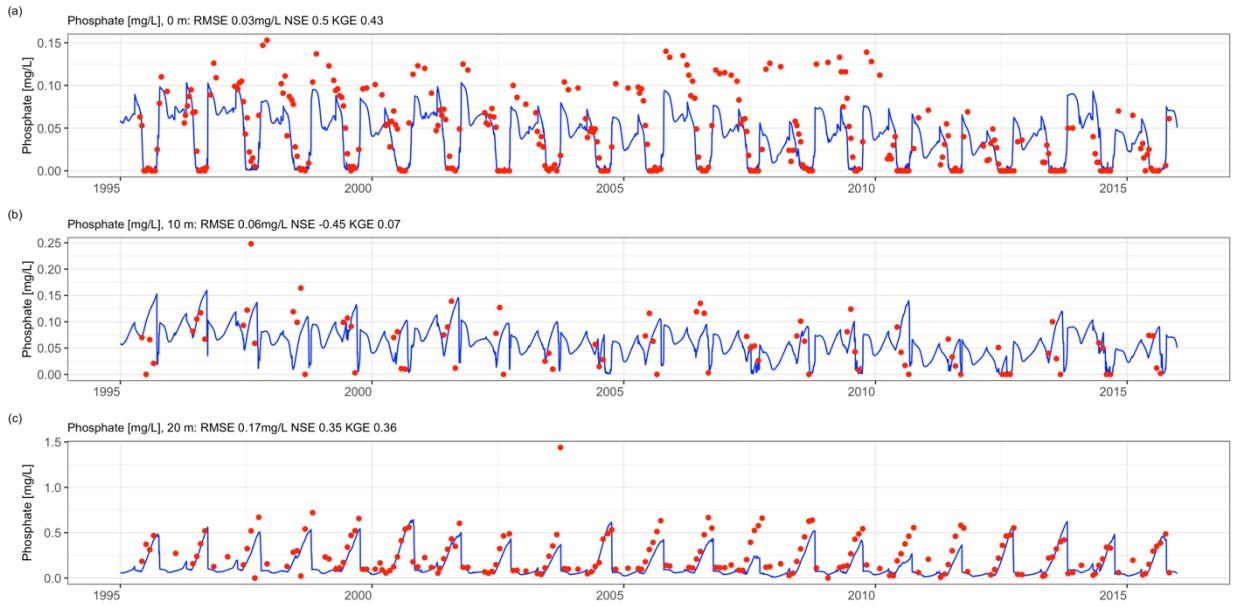

**Figure A7** Time-series comparison between observed (red dots) and modeled phosphate concentrations (blue lines). The fit criteria
root-mean square error (RMSE), Nash-Sutcliffe coefficient of efficiency (NSE) and Kling-Gupta coefficient of efficiency (KGE).

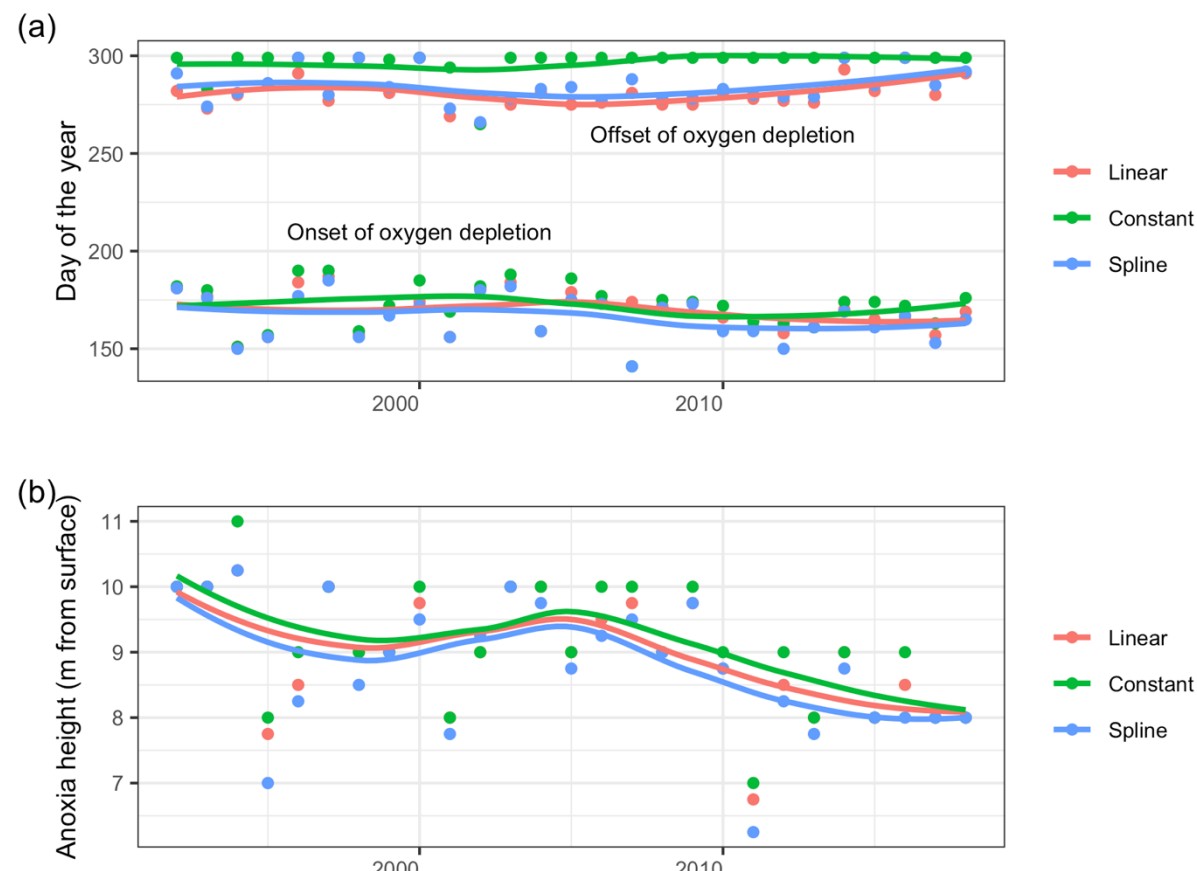

**Figure A8 Observed anoxia onset, offset (a) and height (b) dynamics. The colored lines refer to the interpolation method.**

**Table A1 Model parameters for functional phytoplankton groups**

| Parameter | Description | Cyanobacteria | Diatoms |
|-----------|-------------|---------------|---------|
| P_initial | Initial concentration of phytoplankton (mmol C/m3) | 10 | 8.4 |
| P0 | Minimum concentration of phytoplankton (mmol C/m3) | 0.03 | 0.03 |
| W_p | sedimentation rate (m/d) | 0 | -0.05 |
| Xcc | carbon to chlorophyll ratio (mg C/mg chla) | 50 | 50 |
| R_growth | Phyto max growth rate @20C (/day) | 0.8 | 2.8 |
| fT_Method | Temperature limitation function of growth | CAEDYM style | CAEDYM style |





| | | | |
|---|---|---|---|
| Theta_growth | Arrhenius temperature scaling for growth function (-) | 1.06 | 1.06 |
| T_std | Standard temperature (deg C) | 20 | 15 |
| T_opt | Optimum temperature (deg C) | 28 | 20 |
| T_max | Maximum temperature (deg C | 35 | 32 |
| lightModel | Type of light response function | no photoinhibition | no photoinhibition |
| I_K | Half saturation constant for light limitation of growth (microE/m^2/s) | 25 | 10 |
| KePHY | Specific attenuation coefficient ((mmol C m^3^-1)^1 m^-1) | 0.005 | 0.001 |
| F_pr | Fraction of primary production lost to exudation (-) | 0.005 | 0.002 |
| R_resp | Phytoplankton respiration/metabolic loss rate @ 20 (deg C) | 0.08 | 0.12 |
| Theta_resp | Arrhenius temperature scaling factor for respiration (-) | 1.05 | 1.07 |
| K_fres | Fraction of metabolic loss that is true respiration (-) | 0.6 | 0.6 |
| K_fdom | Fraction of metabolic loss that is DOM (-) | 0.05 | 0.05 |
| simDINUptake | Simulate DIN uptake | True | True |
| simINDynamics | Simulate internal N | Fixed C:N | Dynamic C:N |
| N_0 | Nitrogen concentration below which uptake is 0 (mmol N/m^3) | 0 | 0 |
| K_N | Half-saturation concentration of nitrogen (mmol N/m^3) | 1 | 3.5 |
| X_ncon | Constant internal nitrogen concentration (mmol N/ mmol C) | 0.035 | 0.035 |
| X_nmin | minimum internal nitrogen concentration (mmol N/ mmol C) | 0.06 | 0.077 |
| X_nmax | maximum internal nitrogen concentration (mmol N/ mmol C) | 0.206 | 0.129 |
| R_nuptake | maximum nitrogen uptake rate (mmol N/m^3/d) | 0.068 | 0.13 |
| R_nfix | nitrogen fixation rate (mmol N/mmol C/day) | 0.13 | 0 |
| simDIPUptake | Simulate DIP uptake | True | True |
| simIPDynamics | Simulate internal phosphorus dynamics | Dynamic C:P | Dynamic C:P |





| | | | |
|---|---|---|---|
| P_0 | Phosphorus concentration below which uptake is 0 (mmol P/m^3) | 0 | 0 |
| K_P | Half-saturation concentration of phosphorus (mmol P/m^3) | 0.5 | 0.7 |
| X_pmin | Minimum internal phosphorus concentration (mmol P/mmol C) | 0.0019 | 0.0081 |
| X_pmax | Maximum internal phosphorus concentration (mmol P/mmol C) | 0.0089 | 0.033 |
| R_puptake | Maximum phosphorus uptake rate (mmol P/m^3/d) | 0.0039 | 0.007 |
| simSIUptake | Simulate Si uptake | False | True |
| Si_0 | Silica concentration below which uptake is 0 (mmol Si/m^3 | - | 0 |
| K_Si | Half-saturation concentration of silica (mmol Si /m3) | - | 2.5 |
| X_sicon | Constant internal silica concentration (mmol Si/mmol C) | - | 0.04 |


**Table A2 Calibrated model parameters**

| Parameter | Description | Unit | Default value (Hipsey et al., 2019a, 2019b) | Model value |
|---|---|---|---|---|
| $f_{sw}$ | Solar radiation scaling factor | - | 1.0 | 0.84 |
| $f_{lw}$ | Long-wave radiation scaling factor | - | 1.0 | 0.99 |
| $C_H$ | Bulk aerodynamic coefficient for sensible heat transfer | - | 0.0013 | 0.0014 |
| $T_{z=1,mean}$ | Annual mean temperature of the upper sediment zone | °C | - | 5.07 |
| $T_{z=2,mean}$ | Annual mean temperature of the lower sediment zone | °C | - | 13.47 |
| $F_{max}^{oxy}$ | Max. sediment flux for dissolved oxygen | $\dfrac{mmol}{m^2 d^2}$ | -100.0 | -100.0 |
| $K_{sed}^{oxy}$ | Half-saturation concentration controlling oxygen sediment flux | $\dfrac{mmol}{m^3}$ | 50.0 | 15.0 |





| | | | | |
|---|---|---|---|---|
| $\theta_{sed}^{oxy}$ | Temperature multiplier for oxygen sediment flux | - | 1.0 | 1.08 |
| $R_{mineral}^{dom}$ | Maximum rate of aerobic mineralisation of labile dissolved organic matter at 20 °C | $d^{-1}$ | 0.5 | 0.5 |
| $F_{max}^{dic}$ | Max. sediment flux for dissolved inorganic carbon (DIC) | $\frac{mmol}{m^2 d^2}$ | 4.0 | 250.0 |
| $K_{sed}^{dic}$ | Half-saturation concentration controlling DIC sediment flux | $\frac{mmol}{m^3}$ | 30.0 | 7.0 |
| $\theta_{sed}^{dic}$ | Arrhenius temperature multiplier for DIC sediment flux | - | 1.0 | 1.08 |
| $F_{max}^{rsi}$ | Max. sediment flux for reactive silica | $\frac{mmol}{m^2 d^2}$ | - | 16.42 |
| $K_{sed}^{rsi}$ | Half-saturation concentration controlling silica sediment flux | $\frac{mmol}{m^3}$ | 50.0 | 1.90 |
| $\theta_{sed}^{rsi}$ | Arrhenius temperature multiplier for silica sediment flux | - | 1.0 | 1.08 |
| $R_{nitrif}$ | Maximum rate of nitrification at 20 °C | $d^{-1}$ | 0.1 | 0.03 |
| $R_{denit}$ | Maximum rate of denitrification at 20 °C | $d^{-1}$ | 0.3 | 2.0 |
| $K_{denit}$ | Half-saturation concentration for denitrification | $\frac{mmol}{m^3}$ | 2.0 | 3.0 |
| $F_{max}^{nit}$ | Max. sediment flux for nitrate | | -5.0 | -9.55 |
| $K_{sed}^{nit}$ | Half-saturation concentration controlling nitrate sediment flux | $\frac{mmol}{m^3}$ | 100.0 | 173.13 |
| $F_{max}^{frp}$ | Max. sediment flux for phosphate | $\frac{mmol}{m^2 d^2}$ | - | 0.49 |





| | | | | |
|---|---|---|---|---|
| $K_{sed}^{frp}$ | Half-saturation concentration controlling phosphate sediment flux | $\dfrac{mmol}{m^3}$ | - | 200.0 |
| $\theta_{sed}^{frp}$ | Arrhenius temperature multiplier for phosphate sediment flux | - | 1.0 | 1.0 |

**Table A3 Most parsimonious multiple linear regression model (adjusted $R^2 = 0.88$, p < 0.001) explaining the Anoxic Factor during summer.**

| | Estimate | Std. Error | t value | Pr(>|t|) | Rel. importance [%] |
|---|---|---|---|---|---|
| Intercept | -1.04e-15 | 5.70e-2 | 0.00 | 1.00 | |
| Onset of stratification *(Onset.Strat)* | -6.65e-1 | 1.49e-1 | -4.46 | 1.2e-4 | 22 |
| Schmidt Stability during summer *(Summer.St)* | 5.38e-1 | 1.79e-1 | 2.99 | 5.0e-3 | 17 |
| HBR ratio during summer *(Summer.HBR)* | -4.76e-1 | 2.64e-1 | -1.80 | 0.08 | 13 |
| Gross primary production in the epilimnion (*Epi.GPP*) | 2.34e-1 | 9.35e-2 | 2.50 | 0.01 | 13 |
| Maximum depth of anoxia during summer below surface *(Intensity)* | -1.53e-1 | 7.90e-2 | -1.93 | 0.06 | 11 |
| Birgean Work during summer *(Summer.B)* | -6.44e-1 | 1.72e01 | -3.72 | 0.8e-4 | 11 |
| Water temperature in the hypolimnion at the onset of stratification (*Wtemp.Strat*) | 2.98e-1 | 1.39e-1 | 2.13 | 0.04 | 10 |