# Peer review of "Lake thermal structure drives inter-annual variability in summer anoxia dynamics in a eutrophic lake over 37 years"

_Hydrology and Earth System Sciences, 2020_

## Referee Comment (RC1) · Anonymous Referee #1 · 24 Aug 2020

Thank you very much for the opportunity to review this manuscript describing the internal (physical and biological) and external factors leading to inter-annual variability in the extent anoxia in Lake Mendota. This study uses a combination of three very different types of models to evaluate these various factors. I found this paper very interesting, very well written, and may be very useful to the scientific community. I applaud the authors in using this multi-model approach. However, I think two of the three models have serious flaws that need to be addressed prior to publication.

My main concern is that one of the main takeaways from this paper (internal productivity has very limited effect on interannual differences in anoxia) may not be true. It may

be true that physical mixing drives the overall extent of anoxia (baseline), but I think it is too early to say interannual variability in productivity has little affect. I think two of the models need to be reevaluated prior to making those conclusions:

GLM-AED2. GLM-AED2 simulated the annual progression of anoxia very well, and simulated the importance of stratification driving not only the average changes in DO depletion but also much of the interannual variability in DO associated with stratification. But the model did not capture the interannual variability in surface productivity that may drive the other interannual variability in DO. It clearly could not reproduce the interannual variability in AF. This model had an R2 of only 0.08 and a negative NSE. Part of the problem may be that the model is trying to simulate two very different lakes (one without spiny water fleas and one with them) - all with one set of coefficients (that may not even represent the lake in the first place). Without simulating the big biological change, I am not sure you can get there with this model.

Suggestion: Use GLM-AED2 to only simulate one of the periods, either prior to or after the change in biology. If this does not improve the overall ability to predict AF, then the phytoplankton parameters may have to be adjusted. Without being able to predict most of the variability in AF, I really don't see its use in this paper.

Regression model. I think there are four flaws in the approach used here: 1. Not including loading and in-lake variables that would potentially describe interannual variability in productivity. 2) Including modeling results in a regression analysis. Given that the model does not simulate AF, it appears that using modeling results in the regression may just add noise to the regression or reinforce parameters that are in the model. 3) Using one correlation and one regression to simulate two very different types of lakes, and 4) Using way too many variables in a single multiple regression equation. Even though it appears based on stepwise regression all of the variables are significant, I think it is way over parameterized. Several studies have shown that with regressions using very few observations, many variables can look significant – with each variable coming in to describe one or a few unique observations. A good rule of thumb is to

keep only 1 variable in a multiple regression for each 8-10 observations. So for this regression with 37 (and actually only 28 monitored years) observations, there should only be maybe 3 independent variables.

Suggestions: 1) include variables like actual loading rather than concentrations, include variables that describe inlake productivity (total phosphorus, chlorophyll, Secchi). I am not sure what GPP actually represents. If GPP does describe the changes in chlorophyll, it should be stated. I also do not think it is a good idea to include things describing DO (like maximum height of anoxia) when you are trying to predict AF (this can get to circular reasoning) 2) Only use the 28 actual observations in the correlations and regressions. 3) Look at the correlations for each part of the record (different biological conditions) separately. 4) Stick to correlations and not use regressions. Or if you do look at regressions start simple and add variables only significant when you consider the change in AIC.

My other main concern is that the deductive model seems to say that it is the inlake productivity that is driving the interannual variability in AF, and the other models seem to be saying it is driven by physics and sediment oxygen demand. Maybe with further analysis the models will come to more similar conclusions. If I am wrong with this interpretation, it should be explained better.

Specific Comments: 1. Line-125. Very little information is given on the actual loading. Can these estimates be compared with others?

2. Line 128 – It says here to look at Weng et al. 2020 for a description of the loading regression, but when I look at that paper, I don't see any more than they used a regression, with no statistics either for the monitored sites or the watershed modeling.

3. Line 136 – You mention other data earlier years, who collected that?

4. Line 159 – See comments above about mixing real observations with modeled data.

5. Line 190 – There are lots and lots of parameters in AED, how did you narrow it down

to the ones to start with, you need to start somewhere?

6. Line 215-Can you expect to capture interannual variability in productivity without having the phytoplankton simulate things specific to Lake Mendota?

7. Line 260 – My bet is that anoxia does occur under the ice, but you can't get that from one measurement during the winter.

8. Line 267 – Loads would be better than concentrations. Concentrations generally do not vary much from year to year. If you did really use loads, you should state that. But you should describe this better.

9. Line 273 – See comments above.

10. Line 278- Since Gross primary productivity (GPP) is your only inlake productivity term, you should describe this in more detail. If this is directly related to chlorophyll, maybe this addresses some of my concerns.

11. Line 281 – Consider dropping this whole paragraph.

12. Line 306 – The major conclusion of the deductive model says that water column respiration controls oxygen depletion, yet everything else seems to point to physics. Am I missing something here?? Is water column respiration the cause and physics drives the variability in this? More explanation is needed.

13. Line 322 – Please give the stats for DO. This is really what matters in this paper, especially in the part that varies from year to year.

14. Line 333 – Reorder this paragraph to put the peaks later when you talk about summer.

15. Line 345 – This paragraph could probably be deleted.

16. Line 370 – It says the model captured annual anoxia events. Yes it described the annual development, but right now it does not seem to have any interannual capabilities??

17. Line 374 – See above.

18. Discussion – Need to tie all three model results together better. Right now two say physics and one says productivity.

19. Line 394 – Although I completely agree with you, I am not sure where this comes from given the model results.

20. Line 420 – Again I agree with you, but other than one variable in seven in the regression, I don't know where this comes from. Need to describe this variables importance.

21. Line 425 – Maybe the lack of relations is due to using loading concentrations rather than actual loads. This is what I think the methods say.

22. Line 433 – Is it loads or concentrations. If it is concentrations, that wouldn't surprise me at all. It is not the annual variations in concentrations that drive things, it is the difference in loads.

23. Line 440 - This could be an important point, maybe there is so much oxygen consumption in the bottom, that it dwarfs any water column consumption. But this disagrees with findings of the other models.

24. Line 445. The apparent changes caused by the Spiny water flea may be totally confounding any correlations, regressions, and your GLM-AED2 modeling. You may have to stick to one of the periods to really describe the effects of physics vs internal. Or have two different models.

25. Line 472. Rather than implementing a different type of dynamic model, maybe better capturing change in productivity and clarity, will help in describing the physics.

26. Line 481 – you didn't calibrate the biological parameters, so this should be rewritten.

27. Line 497 – Rather than thinking the deductive model is biased, maybe it is the only approach capturing the effects of the biology.

---

## Referee Comment (RC2) · Chenxi Mi (Referee) · 29 Aug 2020

**General comments:**

This manuscript describes a one-dimensional model (i.e. GLM-AED2) study for Lake Mendota which analyzed its long-term changes of anoxia and the driving factors. As a major result, the model showed good performance in reproducing oxygen dynamics, especially the low oxygen concentration in the hypolimnion, in the lake and based on the statistical analysis, it suggested that the physical structure (e.g. Schmidt Stability, onset of stratification, water temperature in the hypolimnion) had a big influence on the spatial and temporal development of anoxia.

This is an interesting and important study, which could be considered for publication after a minor revision. Although there are quite a few studies analyzing hypolimnetic anoxia for inland waters, most of them draw their conclusion based on the short-term measurements and there is still a need to comprehensively illustrate this phenomenon and mechanisms behind its formation based on long-term database. Based on this prospective, this research fills in a research gap. In my opinion, this paper is well organized and its content, especially the discussion part will improve our understanding about anoxia and its future development under climate warming. Detailed comments are shown below.

**Detailed comments:**

**2.1 Study Site**: It is better to show a topographic map of this lake, as well as the location for the water quality measurements.

**L 115**: 1.How you calibrated the hydrological model?

2. From I know for the historical simulation, the inflow discharge is always drawn from the real measurements, instead of hydrological models. Do you have the measured inflow discharge for Lake Mendota?

**L 125**: How many types of nutrients were included here as the inflow boundary conditions? It is better clarify it here.

**L 133**: I am not sure whether it is appropriate to define the inflow loading as the mean values from the water column. It means that there is no seasonal changes of DIC and silica, which is unrealistic. Could you explain why you set the inflow DIC and silica in this way?

**2.3 Modelling Framework**: Just a recommendation, it may be better to combine 2.3 to 2.7 into one part, since all of such content belongs to the model description.

**L 198**: For water temperature simulation, I supposed the most important parameters should be wind factor and light extinction coefficient. How you defined these two in the model?

**L 293**: How you calculated GPP? It is better to clarify it here.

**L 333**: There existed some negative values for Birgean Work in Figure 5, what is the reason for that?

**L 371**: In Figure 9B, why was the simulated AF represented by dots, instead of box plots as the measured one?

Yours sincerely

Chenxi Mi at Magdeburg

---

## Author Comment (AC1) · 27 Oct 2020

**Referee general comment:**
Thank you very much for the opportunity to review this manuscript describing the internal (physical and biological) and external factors leading to inter-annual variability in the extent anoxia in Lake Mendota. This study uses a combination of three very different types of models to evaluate these various factors. I found this paper very interesting, very well written, and may be very useful to the scientific community. I applaud the authors in using this multi-model approach. However, I think two of the three models have serious flaws that need to be addressed prior to publication.

My main concern is that one of the main takeaways from this paper (internal productivity has very limited effect on interannual differences in anoxia) may not be true. It may be true that physical mixing drives the overall extent of anoxia (baseline), but I think it is too early to say interannual variability in productivity has little affect. I think two of the models need to be reevaluated prior to making those conclusions:

**Referee comment:**
GLM-AED2. GLM-AED2 simulated the annual progression of anoxia very well, and simulated the importance of stratification driving not only the average changes in DO depletion but also much of the interannual variability in DO associated with stratification. But the model did not capture the interannual variability in surface productivity that may drive the other interannual variability in DO. It clearly could not reproduce the interannual variability in AF. This model had an R2 of only 0.08 and a negative NSE. Part of the problem may be that the model is trying to simulate two very different lakes (one without spiny water fleas and one with them) - all with one set of coefficients (that may not even represent the lake in the first place). Without simulating the big biological change, I am not sure you can get there with this model.

**Referee suggestion:** Use GLM-AED2 to only simulate one of the periods, either prior to or after the change in biology. If this does not improve the overall ability to predict AF, then the phytoplankton parameters may have to be adjusted. Without being able to predict most of the variability in AF, I really don't see its use in this paper.

**Author response:**
We are very thankful for this comment by the reviewer, which gives us the chance to discuss the GLM-AED2 performance and hopefully improve the overall manuscript. We make two over-arching points here. The first is that a visible shift in AF occurred in 2010 (Fig. 10b), and this may be explained by changes in the foodweb that affect primary production and organic matter cycling. We have no conclusive evidence of the cause, but the shift is coincident with the invasion of the predacious zooplankton, *Bythotrephes*. We discuss this and have added it to the abstract. The second is that our model reproduces well the ecosystem dynamics prior to 2010, and as the reviewer suggests, the lake is likely in different states, separated by the shift that occurs in 2010. Further, we also acknowledge that the GPP is actually an important driver of the variability in summer anoxia (rel. importance 15 %). The main text was revised accordingly in the abstract and in the discussion:

> L27-32: The summer heat budget, the timing of thermal stratification, and the gross primary production in the epilimnion were the most important predictors of the spatial and temporal extent of summer anoxia periods in Lake Mendota. Inter-annual variability in anoxia was largely driven by physical factors: earlier onset of thermal stratification in combination with a higher vertical stability strongly affected the duration and spatial extent of summer anoxia. A step change upward in summer anoxia in 2010 was unexplained by the GLM-AED2 model. Although the cause remains unknown, possible factors include invasion by the predacious zooplankton, *Bythothrephes longiman*us.

> L441-443: We also acknowledge that a step change in the Anoxic Factor occurred in 2010 and was unexplained by our model. Although the cause remains unknown, the timing was coincident with large increases in the invasive zooplankton, *Bythotrephes* (Walsh et al., 2017).

To the point about the model not capturing variability in surface productivity, we added a new figure to the Appendix: Figure A8 which shows the time-series comparison between observed and modeled DOC concentrations. Here, you can see that the model replicated the overall dynamics of DOC concentrations in three different depths over time, which highlights its ability to replicate net aquatic production in the surface layer and its contribution to dissolved organic matter. Fig. 9a of the main text showed that the model overestimates surface dissolved oxygen concentration. This overestimation must have a concomitant increase in organic matter as a consequence of photosynthesis, and in this case is particulate organic matter (POM). Considering our proxy for phytoplankton biomass is well predicted (Fig. 5), this suggests our over-estimate of primary production results in

increase in POM that is exported from the epilimnion to the hypolimnion. Unfortunately, we do not have observed POM to calibrate this part of the model, but we feel it is likely that our model has overestimated the contribution of primary production to hypolimnetic organic matter and subsequent oxygen depletion. This underlies our conclusion that primary production may be less important to inter-annual variability than physical factors. We added these sentences to the main text to state this:

> L481-487: Although the model replicated well the long-term DOC dynamics (Appendix Figure A8), it also overestimated surface layer dissolved oxygen concentrations compared to the observed data. This overestimation must have a concomitant increase in organic matter as a consequence of photosynthesis, and in this case in POC. Considering our proxy for the dynamics of phytoplankton biomass is reasonably well predicted (Fig. 5), this suggests our over-estimate of primary production results in increase in POC that is exported from the epilimnion to the hypolimnion. Unfortunately, we do not have observed POC to calibrate this part of the model, but we feel it is likely that our model has overestimated the contribution of primary production to hypolimnetic organic matter and subsequent oxygen depletion.

[Figure]

**Figure A8 Time-series comparison between observed (red dots) and modeled dissolved organic carbon concentrations (blue lines). The fit criteria root-mean square error (RMSE), Nash-Sutcliffe coefficient of efficiency (NSE) and Kling-Gupta coefficient of efficiency (KGE).**

Regarding the capture of interannual anoxia dynamics: Yes, it seems there was a shift in the ecosystem happening beginning in 2010 with higher annual Anoxic Factors. We changed Figure 10 to also show the comparison between simulated Anoxic Factor and the observed data for the periods pre-2010 and post-2010. We also recalculated goodness of fit separately for the two time periods. For the total time period (Fig 10b, 1992-2015 when observed data was available) the model achieved an RMSE of 7.12 d, NSE of -0.22, KGE of 0.26 and r of 0.28 showing that on average it was a week off in replicating the Anoxic Factor, but the KGE and r values proved that the general dynamics and interannual variability could be replicated. When comparing with the pre-2010 period (Fig 10c), the model achieved an RMSE of 6.79 d, an NSE of -0.25, an KGE of 0.44 and r of 0.45, which highlights the model's ability to replicate anoxia dynamics in this period (please note that the model was calibrated for the period 2005-2015 which proves, at least in our opinion, the success of the calibration if there indeed was an ecosystem shift). When comparing with the post-2010 period (Fig 10d), the model achieved an RMSE of 8.04 d, an NSE of -31.99, an KGE of 0.21 and r of 0.62. Here, the model is biased as the observed Anoxic Factor is higher in all years except 2013. Still, the interannual variability expressed by the correlation coefficient r was captured very well by the model. The p-value for the pre-2010 period of the correlation coefficient was p=0.0591. For the post-2010 period, the p-value = 0.19, reducing our confidence in the model for this shorter time period. The visual inspection of

these plots (10c and 10d) highlights that they represent different ecosystem states, as there is step-change in the Anoxic Factor starting in 2010.

[Figure]

**Figure 1 Comparison of observed to modeled dissolved oxygen concentrations and ecosystem response.** A Contour plot of observed (upper figure, white dots mark sample events) and simulated dissolved oxygen concentrations. B Comparison of

**simulated Anoxic Factor (red dots) against interpolated range of Anoxic Factor derived from observed data (box-whisker plots) over the period 1979 to 2018. C Comparison of simulated Anoxic Factor (red dots) against interpolated range of Anoxic Factor derived from observed data (box-whisker plots) over the period 1992 to 2009. B Comparison of simulated Anoxic Factor (red dots) against interpolated range of Anoxic Factor derived from observed data (box-whisker plots) over the period 2010 to 2015.**

Therefore, we focused our regression analysis on the pre-2010 period. First, we inspected if the distributions of the observed and modeled Anoxic Factors were similar by investigating the null hypothesis that they are identical populations as determined by the Wilcoxon test (see attached figure below). This test achieved a non-significant p-value of 0.13, indicating strong overlap in populations, and therefore are comparable. We added this figure as Figure A9 to the Appendix A of the manuscript. Also, a similar comparison of the Anoxic Factors for the post-2010 period revealed that observed and modeled distributions were significantly different with a p-value of 0.032. This effectively highlights that we can talk about "two different lakes here". We added these sentences to the main text in the results and in the discussion:

> L412-421: The simulated Anoxic Factor over the total time period averaged 56.7 ± 5.2 days with an RMSE of 7 days, an NSE of -0.22, and an KGE of 0.26 (correlation coefficient r = 0.28). The model's underestimation of the recent positive trend of Anoxic Factors starting in 2010 was investigated by quantifying the fits during two periods: 1992-2009 (Figure 10C) and 2010-2005 (Figure 10D). In the pre-2010 period (1992-2009), the model achieved an RMSE of 6.79 days, an NSE of -0.25, an KGE of 0.44 and r of 0.45 for Anoxic Factor predictions. In the post-2010 period (2010-2015), the model achieved an RMSE of 8.04 days, an NSE of -31.99, an KGE of 0.21 and r of 0.62. A subsequent Wilcoxon signed-rank test highlighted, that the observed average and modelled Anoxic Factors from the pre-2010 period showed no significant differences between the two distributions, suggesting they belong to the same population (p-value = 0.13, Appendix Figure A9), whereas the distributions of observed mean Anoxic Factors and modeled ones after 2010 were significantly different (p-value = 0.032, Appendix Figure A9).

[Figure]

**Figure A9 Box-whisker plots of observed to modeled Anoxic Factor for (a) the period 1992-2009 and (b) for the period 2010-2015.**

We discussed novel insights into these two distinct periods by expanding this paragraph

> L498-518: The model replicated the maximum anoxia event in 1998 but struggled to replicate the minimum in 2002. The discrepancies of 5-10 days between the simulated and observed range of the Anoxic Factor beginning in 2010 are related to an increased spatial as well as temporal extent of summer anoxia (Appendix Figure A10), which was not captured by the model. This was highlighted by the statistical analysis of the pre-2010 (1992-2009) and post-2010 (2010-2015) Anoxic Factors. Prior to 2010, there were no significant differences between observed and modeled distributions (p=0.13); whereas,

after 2010, the observed distribution was significantly higher than the modeled distribution (p=0.032) (Appendix Figure A9). For simplicity and due to limitations in Lake Mendota monitoring data post-2010, we focused the regression analysis of the Anoxic Factor in this study only on the pre-2010 period. The change in Anoxic Factor post-2010 may be due to an ecosystem shift in Lake Mendota that began in 2009, when the invasive spiny water flea (*Bythothrephes longimanus*) was detected in surprisingly high densities in the lake (Walsh et al., 2016b, 2018). Spiny water flea effectively became the dominant *Daphnia* grazer, causing historically low *Daphnia* biomass in 2010, 2014 and 2015 (Walsh et al., 2016a) and reducing water clarity. The spiny water flea may have increased organic matter supply to the hypolimnion by grazing down certain phytoplankton. Mendota's Daphnia population historically consisted of *Daphnia pulicaria* and the smaller-bodied *Daphnia galeata mendotae,* who compete differently with spiny water flea*. D. mendotae* biomass increased in spring after the spiny water flea invasion (Walsh et al., 2017), grazing on phytoplankton and probably accelerating organic matter mineralization before stratification onset. This could be one potential cause that contributed to the increase in hypolimnetic oxygen depletion after 2010. Our GLM-AED2 model could not replicate this food web change, and subsequent shift in anoxia dynamics, due to limitations of the numerical model, i.e., GLM-AED2 had constant ecological parameters over the entire modeling period and did not have zooplankton dynamics instantiated. We envision future monitoring and modeling studies that focus entirely on ecosystem differences and shifts between the pre-2010 and post-2010 periods of Lake Mendota.

Further, by analyzing the autocorrelation function (ACF) of the observed mean Anoxic Factors and the modeled ones (see figure below), we concluded that there is no autocorrelation between annual Anoxic Factors. It may be the case that the interannual variation in the Anoxic Factor (investigated by ACF) is effectively random, which does not mean that the Anoxic Factor is necessarily random, but that the variation in external drivers may be random. Still, our model's simulated Anoxic Factors are from the same distribution as the observed mean values highlighting the model's ability to capture the overall distribution of anoxia. Further, the fit metrics (highlighted in revised Figure 10) show that the model can capture inter-annual variability significantly prior to 2010, even if the average value is off by about a week.

[Figure]

We therefore followed the reviewer's suggestion to only include the pre-2010 period for the regression analysis, which is discussed in the next comment and response block:

> L314-317: Only model output and model driver data from the period 1980-2009 were used in the regression analysis. The first year, 1979, was dropped from the investigations due to a lack of prior winter information. The years 2010-2015 were dropped due to an apparent ecosystem shift (see Section '3.4 Oxygen Dynamics').

**Referee comment:**
Regression model. I think there are four flaws in the approach used here: 1. Not including loading and in-lake variables that would potentially describe interannual variability in productivity. 2) Including modeling results in a regression analysis. Given that the model does not simulate AF, it appears that using modeling results in the regression may just add noise to the regression or reinforce parameters that are in the model. 3) Using one correlation and one regression to simulate two very different types of lakes, and 4) Using way too many variables in a single multiple regression equation. Even though it appears based on stepwise regression all of the variables are significant, I think it is way over parameterized. Several studies have shown that with regressions using very few observations, many variables can look significant – with each variable coming in to describe one or a few unique observations. A good rule of thumb is to keep only 1 variable in a multiple regression for each 8-10 observations. So for this regression with 37 (and actually only 28 monitored years) observations, there should only be maybe 3 independent variables.

**Referee suggestion:** 1) include variables like actual loading rather than concentrations, include variables that describe inlake productivity (total phosphorus, chlorophyll, Secchi). I am not sure what GPP actually represents. If GPP does describe the changes in chlorophyll, it should be stated. I also do not think it is a good idea to include things describing DO (like maximum height of anoxia) when you are trying to predict AF (this can get to circular reasoning) 2) Only use the 28 actual observations in the correlations and regressions. 3) Look at the correlations for each part of the record (different biological conditions) separately. 4) Stick to correlations and not use regressions. Or if you do look at regressions start simple and add variables only significant when you consider the change in AIC.

**Author response:**
Thank you for your very thoughtful explanation of the regression analysis' flaws and your very helpful suggestions how to overcome these.

1)  We changed the inflow variables, total phosphorus inflow concentration and total nitrogen inflow concentration (both in g per m2), to total phosphorus inflow loading and total nitrogen inflow loading (both now in g per d per m2 of lake area). These loading variables were included in the model to assess the importance of external hydrological drivers for the extent of anoxia. To capture in-lake productivity variables, our regression included the cumulative gross primary production in the surface and bottom lake that represent the total sum of photosynthesis, hence expressed as carbon uptake, of each functional phytoplankton group, and scales directly with in-lake Chl-a concentrations. Further, our regression also includes the temporal change of dissolved as well as particulate organic carbon in the bottom layer from stratification onset to fall mixing onset. To make it clearer what GPP represents, we added this sentence to the main text:

    L300-308: Here, GPP represents the sum of all functional phytoplankton group's photosynthesis rates parameterized as the total carbon uptake:

    $$f_{uptake}^{PHY_C} = R_{growth}^{PHY}(1 - k_{pr}^{PHY}) \, \phi_{temp}^{PHY}(T) \, \phi_{stress}^{PHY}(X) \, min\{\phi_{light}^{PHY}(I) \, \phi_N^{PHY}(NO_3, NH_4 PHY_N) \, \phi_P^{PHY}(PO_4, PHY_P) \, \phi_{Si}^{PHY}(Rsi)\}[PHY] \quad (7)$$

    where the carbon uptake $f_{uptake}^{PHY_C}$ of an individual group *PHY* depends on the growth rate $R_{growth}^{PHY}$, the photorespiratory loss $(1 - k_{pr}^{PHY})$, temperature scaling $\phi_{temp}^{PHY}(T)$, metabolic stress $\phi_{stress}^{PHY}(X)$, and a minimum function taking into account limitations by light $\phi_{light}^{PHY}(I)$, nitrogen $\phi_N^{PHY}(NO_3, NH_4 PHY_N)$, phosphorus $\phi_P^{PHY}(PO_4, PHY_P)$ and silica $\phi_{Si}^{PHY}(Rsi)\}$ (Hipsey et al., 2017; adapted from Hipsey and Hamilton, 2008). As the GPP is the main model output variable for phytoplankton dynamics, it scales directly with biomass and Chl-a concentrations.

    Following the reviewer's suggestion, we removed the maximum height of anoxia in the regression analysis. The variable was removed from all paragraphs (2.3.4 Regression Model, Table 1)

2)  For the regression we only used modeled results and no actual observed data. This was done to identify internal connections in the numerical model and its mathematical equations. Similar analyses of modeled output and model driver data were done in Farrell et al., 2020; Snortheim et al., 2017; Ward et al., 2020. We added these sentences to the Methods section:

    L279-282: All candidate predictors were either modeled output or boundary data for the model. This enabled the regression analysis to identify internal connections in the numerical model itself (similar

analyses of modeled output and driver data were done in Snortheim et al., 2017; Ward et al., 2020; Weng et al., 2020).

3) Following our reasoning in the first comment and response section, and the suggestions by the reviewer we revised our regression analysis by only using model data from 1980-2009. This excludes the first year as warm-up period and the post-2010 period due to different ecosystem conditions (probably spiny water flea invasion). We added additional discussions regarding the ecosystem shift (see response to first comment).

4) Following the suggestion of the reviewer, we re-did the regression analysis with 21 candidate predictors using model output and model drivers from 1980-2009 (we removed the anoxia height from the sediment) using the Boruta algorithm (random forest classifier). This analysis identified 10 variables as important. Subsequently, we did a step-wise analysis of the AIC of each model. This resulted in the identification of seven predictors: HBR ratio during spring, HBR ratio during summer, Birgean Work in spring, epilimnetic GPP, Schmidt Stability in summer, Birgean Work in summer, and onset date of stratification. The AICs of each model with any of these variables removed did not result in significant changes (this table was added to the manuscript as Table A3):

**Table A3 Step-wise model-selection by removing predictors of the multiple linear regression model using seven predictors.**

| Predictor | AIC |
|---|---|
| HBR ratio during spring *(Spring.HBR)* | -61.820 |
| HBR ratio during summer *(Summer.HBR)* | -60.529 |
| Birgean Work during spring *(Spring.Birgean)* | -60.189 |
| Gross primary production in the epilimnion (*Epi.GPP*) | -58.952 |
| Schmidt Stability during summer *(Summer.St)* | -51.829 |
| Birgean Work during summer *(Summer.B)* | -50.848 |
| Onset of stratification *(Onset.Strat)* | -42.900 |

We reduced the final model to only three predictors (as the reviewer suggested) including onset date of stratification, Schmidt Stability in summer (as the AIC was similar to Birgean but the concept of Schmidt Stability is more generally known) and epilimnetic GPP. The text in "2.3.4 Regression Model" was accordingly changed to:

L321-330: This multiple linear regression model to predict Anoxic Factor included seven variables: HBR ratio during spring, HBR ratio during summer, Birgean Work in spring, Schmidt Stability in spring, epilimnetic GPP, Schmidt Stability in summer, Birgean Work in summer, and onset date of stratification. We reduced the complexity of the final multiple linear regression model to only three predictors of Anoxic Factor: onset date of stratification, Schmidt Stability in summer, and epilimnetic GPP. Schmidt Stability was included instead of Birgean Work as the resulting AIC of both models were similar, but the concept of Schmidt Stability is more commonly used in the limnological research community (Appendix Table A3). The final multiple linear regression model was configured as (scaled predictors, adjusted $R^2$ = 0.84, p < 0.001 Appendix Table A4).

$$\hat{y} = 0.24 Epi.GPP + 0.54 Summer.St - 0.46 Onset.Strat - 5.44 * 10^{-17} + \hat{\epsilon}, \tag{8}$$

where $\hat{\epsilon} \, N(0,38^2)$.

The results text in "3.5 Regression Model" was changed to:

L423-430: We included in total 3 predictors in our final multiple linear regression which were deemed important by the Boruta algorithm and stepwise linear model investigations using AIC for the period 1980-2009: Schmidt Stability during summer, the onset date of stratification, and gross primary production in the epilimnion (Appendix Table A4).

The linear model showed a good agreement between simulated and predicted Anoxic Factor (Figure 11 A, Appendix Table A4). The Anoxic Factor was positively correlated to the summer Schmidt Stability (r = 0.72, Figure 11 B) and the gross primary production in the epilimnion (r = 0.48). It was negatively correlated to the onset of stratification (r = -0.78, Figure 11 B).

We changed Appendix Table A3 (formerly A2) and Figure 11 accordingly:

**Table A3 Most parsimonious multiple linear regression model (adjusted $R^2 = 0.84$, p < 0.001) explaining the summer Anoxic Factor.**

|  | Estimate | Std. Error | t value | Pr(>\|t\|) | Rel. importance [%] |
|---|---|---|---|---|---|
| Intercept | -1.04e-15 | 5.70e-2 | 0.00 | 1.00 |  |
| Schmidt Stability during summer *(Summer.St)* | 5.386e-1 | 7.920e-2 | 6.800 | 3.23e-7 | 43 |
| Onset of stratification *(Onset.Strat)* | -4.581-1 | 9.006e-2 | -5.086 | 2.68e-5 | 42 |
| Gross primary production in the epilimnion (*Epi.GPP*) | 2.436e-1 | 8.327e-2 | 2.926 | 0.00704 | 15 |

[Figure]

(a)

(b)

[Figure]

**Figure 2 Predicted against simulated summer Anoxic Factor. A Linear model with a prediction which was done using a multiple linear regression model of the form:** $\hat{y} = 0.24 Epi.GPP + 0.54 Summer.St - 0.46 Onset.Strat - 5.44 * 10^{-17} + \hat{e}$**, where** $\hat{e} \ N(0, 38^2)$**. The red lines represent confidence intervals. B Correlogram of the input data using Pearson correlation coefficients**

We changed the following sentences in the main text to reflect these changes:

L434-438: The Schmidt Stability during summer (rel. importance of 43 %) as well as the timing of stratification (rel. importance of 42 %) all influence Anoxic Factor, and are all driven mainly by atmospheric drivers and heat convection throughout the water column. The most important predictor of Anoxic Factor directly related to biological processes is gross primary production in the epilimnion (rel. importance of 15 %), Appendix Table A4).

L596-599: Physical metrics – summer Schmidt Stability and onset date of stratification – were the most important predictors driving the summer Anoxic Factor. Although the gross primary production was still influential in affecting year-to-year variability of hypolimnetic anoxia, biological control over the Anoxic Factor was limited in our study period.

**Referee comment:**
My other main concern is that the deductive model seems to say that it is the inlake productivity that is driving the interannual variability in AF, and the other models seem to be saying it is driven by physics and sediment oxygen demand. Maybe with further analysis the models will come to more similar conclusions. If I am wrong with this interpretation, it should be explained better.

**Author response:**
The deductive model itself can only determine between two sources of depletion, either a volumetric one or an area sink. It cannot distinguish between biological or physical drivers of these depletion causes. Although the deductive model states that the volumetric sink is higher than the area sink, this is only of importance for the in-lake biological drivers (as the area sink depends on in-situ biogeochemical conditions). In the manuscript we state that the anoxia variability over a summer season is mainly driven by changes in the physical drivers, whereas we acknowledge that oxygen depletion itself (as shown in the regression model) is a function of biological and chemical activity. The deductive model itself does not consider any physical drivers, even diffusion is neglected. We added these lines to the main text to clarify our message:

> L529-532: We note that the simple deductive model itself can only differentiate between two sources of depletion and neglects any physical transport drivers of oxygen, e.g., diffusion. Therefore, the results of the deductive model only add direct information to the actual depletion process of dissolved oxygen, but not of the dominant drivers.

**Referee comment:**
1. Line-125. Very little information is given on the actual loading. Can these estimates be compared with others?

**Author response:**
Thank you. We compared our loadings with literature values, especially regarding phosphorus. Previous estimates range from about 15-67 t of total phosphorus (TP) per year (Kara 2011). Our estimates are at the higher end of this range. There is a concern that previous estimates did not fully account for loads of adsorbed phosphorus (hence, phosphate bound on sediment), because of the importance of extreme storm events on particulate loads (Carpenter 2017). To accommodate for a potential underestimation of TP loads, we added to the inflow boundary condition the adsorbed phosphate variables, which was set roughly equal in magnitude to non-adsorbed phosphorus. This puts our estimates of total P load near the upper range of previous estimates. Bennett (1999) estimated the long-term average annual TP input with 34 t P. Our Yahara inflow had an average annual TP load of about 25.3 t/y and ranged between 2.69 to 73.09 t/y over the period 1979-2015. Due to the use of a hydrological model, our inflows accounted for a closed water balance and included near-lake groundwater/spring inflows. Our average annual load of 25.3 t/y is slightly higher than the loadings by of Lathrop (2009). We added these lines to the main text:

> L136-144: To provide information regarding adsorbed soluble reactive phosphate, we doubled measured total phosphorus (TP) concentrations and applied specific ratios to individual phosphorus forms (Farrell et al., 2020; Snortheim et al., 2017; Weng et al., 2020). This put our estimates of TP near the upper range of previous load estimates. Bennett et al., (1999) estimated the long-term average annual TP load to be about 34 t, whereas our average annual TP load (with adsorbed phosphate) was about 50.6 t and ranged between 5.3 to 146.1 t (1979-2015). Our average annual TP load (without adsorbed phosphate) was about 25.3 t and ranged between 2.7 to 73.1 t (1979-2015), which is similar to previous estimates between 15 to 67 t (Kara et al., 2012). By doubling our TP by adding adsorbed phosphate, we accommodate a potential TP load underestimation due to the importance of extreme storm events on particulate loads (Carpenter et al., 2018).

Further, we checked our derived annual TP loadings using the Vollenweider model by assuming winter TP concentrations, $TP_{lake}$, of 140 ug/L, a residence time, RT, of 4 years, P retention, $\sigma$, of 0.7, and a mean depth, $z_{mean}$, of 12.8 m:

$$TP_{lake} = \frac{L}{z_{mean}\left(\frac{1}{RT} + \sigma\right)}$$

$$L = 0.14 \, g/m3 \, (12.8 \, m \, (0.25 \, y^{-1} + 0.7 \, y^{-1})) = 1.70 \, g/m2/y$$

By multiplying L with the lake area of Lake Mendota (approx. 39.61 km2), the Vollenweider model quantifies the annual load for steady-state conditions with 67 t/y, which is slightly above our average annual TP load (with adsorbed phosphate) of 50.6 t/y.

**Referee comment:**
2. Line 128 – It says here to look at Weng et al. 2020 for a description of the loading regression, but when I look at that paper, I don't see any more than they used a regression, with no statistics either for the monitored sites or the watershed modeling.

**Author response:**
Thank you for pointing this out. Yes, there are no previous publications describing the regression fit analysis. We described in the previous response that our TP loads were near the upper range of previous estimates due to our addition of adsorbed phosphate (due to extreme storm events and land erosion). For the regression analysis between discharges and nutrient concentrations, we used the state-of-the-art loadflex R-package (https://github.com/USGS-R/loadflex, Appling et al., 2015). As monitored TP estimates are rare, a comprehensive statistical analysis is challenging. The attached figure visualizes the fit for 8 years (2008-2015), which was satisfactory for most years. USGS monitoring began Oct 2008, so a comparison cannot be made for the entire year. Overall, our model tended to overestimate nutrient loads into the lake.

| Year | LOADEST.kg | USGSLoad.kg |
|------|-----------|-------------|
| 1996 | 10848 | NA |
| 1997 | 8827 | NA |
| 1998 | 14289 | NA |
| 1999 | 12268 | NA |
| 2000 | 15558 | NA |
| 2001 | 14384 | NA |
| 2002 | 10571 | NA |
| 2003 | 6703 | NA |
| 2004 | 15039 | NA |
| 2005 | 5139 | NA |
| 2006 | 9515 | NA |
| 2007 | 12412 | NA |
| 2008 | 25341 | 1478 |
| 2009 | 15793 | 22790 |
| 2010 | 19075 | 11361 |
| 2011 | 10792 | 10398 |
| 2012 | 4122 | 4255 |
| 2013 | 14013 | 12767 |
| 2014 | 12205 | 9988 |
| 2015 | 8007 | 5293 |

[Figure]

Phosphorus load at USGS 05427850 YAHARA RIVER AT STATE HIGHWAY 113

**Referee comment:**
3. Line 136 – You mention other data earlier years, who collected that?
**Author response:**
The additional data points were measured by Patricia Soranno for her thesis. We added that information in the sentence:

> L149: The dissolved oxygen data set was complemented with historical measured dissolved oxygen data from 1992 to 1994.

And we acknowledged her in the Acknowledgement section "We are thankful for supplementary dissolved oxygen field data from 1992-1994 by Patricia Soranno." Her data does not officially belong to the NTL-LTER monitoring data set of Lake Mendota, but it gave us valuable early spring-summer information regarding oxygen dynamics.

**Referee comment:**
4. Line 159 – See comments above about mixing real observations with modeled data. 5. Line 190 – There are lots and lots of parameters in AED, how did you narrow it down to the ones to start with, you need to start somewhere?
**Author response:**
We used the Morris Sensitivity Method to identify crucial parameters for the calibration. For this analysis we included the main model parameters regarding sediment flux and in-water biogeochemical reactions, mainly, of the main nutrient modules: oxygen, carbon, silica, nitrogen and phosphate. For the initial values, we chose starting values either from the AED2 webpage (https://aed.see.uwa.edu.au/research/models/aed/modules.html, default values, or values inside the typical range) or from previous modeling work on Lake Mendota (see Snortheim et al. 2017). We added this sentence to the main text for clarification:

> L231: Initial model parameter values were taken from default parameter values and ranges, as well as literature values (Hipsey et al., 2017; Snortheim et al., 2017).

**Referee comment:**
6. Line 215-Can you expect to capture interannual variability in productivity without having the phytoplankton simulate things specific to Lake Mendota?
**Author response:**
This is a good point, thank you for raising this. Although we did not calibrate the functional phytoplankton groups specifically to Lake Mendota, we still checked simulated Chl-a and Secchi depth values, as well as timings of phytoplankton bloom peaks. In general, the model did replicate the seasonal succession well. We've attached the

following figure that compares the observed to modeled Secchi depths for the reviewer to inspect. The summer Secchi depths from the model are similar to the observed ones, highlighting that the ecosystem dynamics during anoxia are similar. The gray boxes highlight the time period from day of the year 150 to day of the year 180 (June to end of August): for the majority of years the model can replicate the Secchi depth dynamics of the June period., whereas generally it underestimates the initial summer Secchi depth.

[Figure]

**Referee comment:**

7. Line 260 – My bet is that anoxia does occur under the ice, but you can't get that from one measurement during the winter.

**Author response:**

Yes, measurements and the monitoring have shown that there is anoxia under the ice in Lake Mendota, but it varies a lot. We agree that we cannot determine the full anoxia extent under ice with only one or two measurements per season. We changed sentence the sentence accordingly to:

L276: We quantified the seasonal Anoxic Factor only for the summer season.

**Referee comment:**
8. Line 267 – Loads would be better than concentrations. Concentrations generally do not vary much from year to year. If you did really use loads, you should state that. But you should describe this better.
**Author response:**
We changed the inflow variables, total phosphorus inflow concentration and total nitrogen inflow concentration (both in g per m2), to total phosphorus inflow loading and total nitrogen inflow loading (both now in g per d per m2). These loading variables were included in the model to assess the importance of external hydrological drivers for the extent of anoxia. We changed the information in Table 1 accordingly:

| Total phosphorus inflow loading | Winter/spring/summer of year n-1 and n | Extracted from driver data | g P per day per m$^2$ |
|---|---|---|---|
| Total nitrogen inflow loading | Winter/spring/summer of year n-1 and n | Extracted from driver data | g N per day per m$^2$ |

**Referee comment:**
9. Line 273 – See comments above.
**Author response:**
Please see our response above.

**Referee comment:**
10. Line 278- Since Gross primary productivity (GPP) is your only in-lake productivity term, you should describe this in more detail. If this is directly related to chlorophyll, maybe this addresses some of my concerns.
**Author response:**
Thank for raising this point. GPP (gross primary productivity) in the lake is the cumulative photosynthesis, represented by cumulative carbon uptake per time step, of all functional phytoplankton groups. Therefore, it scales directly with the simulated Chl-a output. We clarified this in the main text:

L300-308: Here, GPP represents the sum of all functional phytoplankton group's photosynthesis rates parameterized as the total carbon uptake:
$$f_{uptake}^{PHY_C} = R_{growth}^{PHY}(1 - k_{pr}^{PHY}) \, \phi_{temp}^{PHY}(T) \, \phi_{stress}^{PHY}(X) \, min\{\phi_{light}^{PHY}(I) \, \phi_N^{PHY}(NO_3, NH_4PHY_N) \, \phi_P^{PHY}(PO_4, PHY_P) \, \phi_{Si}^{PHY}(Rsi)\}[PHY]$$
(7)
where the carbon uptake $f_{uptake}^{PHY_C}$ of an individual group *PHY* depends on the growth rate $R_{growth}^{PHY}$, the photorespiratory loss $(1 - k_{pr}^{PHY})$, temperature scaling $\phi_{temp}^{PHY}(T)$, metabolic stress $\phi_{stress}^{PHY}(X)$, and a minimum function taking into account limitations by light $\phi_{light}^{PHY}(I)$, nitrogen $\phi_N^{PHY}(NO_3, NH_4PHY_N)$, phosphorus $\phi_P^{PHY}(PO_4, PHY_P)$ and silica $\phi_{Si}^{PHY}(Rsi)\}$ (Hipsey et al., 2017; adapted from Hipsey and Hamilton, 2008). As the GPP is the main model output variable for phytoplankton dynamics, it scales directly with biomass and Chl-a concentrations.

**Referee comment:**
11. Line 281 – Consider dropping this whole paragraph.
**Author response:**
Although we understand the reasoning behind dropping this paragraph as the same results could probably be achieved by either starting with a simple linear regression model and extending it, or by step-wise analysis of AIC, we decided to keep the Boruta algorithm and analysis in the manuscript. This method allows us to analyze 21

potential predictors in a comprehensive framework before reducing the final number of important predictors by step-wise analysis.

**Referee comment:**
12. Line 306 – The major conclusion of the deductive model says that water column respiration controls oxygen depletion, yet everything else seems to point to physics. Am I missing something here?? Is water column respiration the cause and physics drives the variability in this? More explanation is needed.
**Author response:**
Please see our response above regarding the limitations of the deductive models and its incapability to acknowledge physical drivers.

**Referee comment:**
13. Line 322 – Please give the stats for DO. This is really what matters in this paper, especially in the part that varies from year to year.
**Author response:**
We agree, thank you. We added these sentences to the main text:

> L356-360: The simulated dissolved oxygen concentrations in the whole water column achieved an RMSE of 3.22 mg L-1, an NSE of 0.56, and an KGE of 0.77. Here, the average fits were better in the surface layer (RMSE of 2.77 mg L-1) compared to the bottom layers (RMSE of 3.31 mg L-1), whereas the temporal dynamics (as expressed in NSE and KGE) were slightly better in the bottom layer (an NSE of 0.64, KGE of 0.81) compared to the surface layer (NSE of -0.36, KGE of 0.47).

Further, the discussion of the oxygen fit has its own subparagraph in "3.4 Oxygen Dynamics" where we state that:

> L405-409: Dissolved oxygen dynamics, including the spatial extent of oxygen depletion in the water column, and the timing of summer anoxia periods, were replicated by the GLM-AED model (Figure 9A-B); although the model overestimated spring and summer time surface oxygen concentrations due to a higher net ecosystem production. The depth-averaged fit criteria of dissolved oxygen concentrations were similar to a recent study from Farrell et al. (2020) in which the RMSE were 1.88 mg/L and 2.49 mg/L in the epilimnion and hypolimnion, respectively, of a GLM-AED model calibrated for Lake Mendota.

**Referee comment:**
14. Line 333 – Reorder this paragraph to put the peaks later when you talk about summer.
**Author response:**
We agree. We moved the sentence to a later paragraph and combined it with the description of the annual course of Schmidt Stability:

> L395: Schmidt Stability peaked on average in August at approx. 720 J m-2 (Figure 6), followed by a peak in the Birgean Work at approx. 1250 J m-2.

**Referee comment:**
15. Line 345 – This paragraph could probably be deleted.
**Author response:**
As the main take-away message of our manuscript is related to physical drivers, we decided to keep this short paragraph describing the deep-water stagnancy in the manuscript. By comparing the additional energy demands of Lake Mendota with other similar sized lakes, the reader gets valuable information regarding the lake's energy budget, and potential conclusions to the anoxia drivers of similar lake systems.

**Referee comment:**
16. Line 370 – It says the model captured annual anoxia events. Yes it described the annual development, but right now it does not seem to have any interannual capabilities??
**Author response:**
We quantified the correlation coefficient for the Anoxic Factor with r = 0.28 (total period), r = 0.45 (pre-2010) and r = 0.62 (post-2010), see also Figure 10. Especially for the pre-2010 period the p-value of the correlation coefficient was p=0.0591, which was slightly above the significance level. Overall, this highlights the model's overall ability to predict interannual changes and dynamics.

**Referee comment:**
17. Line 374 – See above.
**Author response:**
See response above.

**Referee comment:**
18. Discussion – Need to tie all three model results together better. Right now two say physics and one says productivity.
**Author response:**
We disagree that two models point to physical drivers and one to biological ones. The deductive model distinguished the main oxygen consumption as either being a volumetric or an area sink term. This information was used to set up the sediment oxygen demand in the GLM-AED2 model. The results of the calibrated GLM-AED2 model were then used in a regression analysis to identify internal connections of the numerical model and its mathematical equations. This confirmed that in the process-based GLM-AED2 model three variables were important predictors of anoxia and its interannual variability. The deductive model itself does not consider any physical drivers (see responses above please).

**Referee comment:**
19. Line 394 – Although I completely agree with you, I am not sure where this comes from given the model results.
**Author response:**
The statement regarding that […] Biology matters but its interannual dynamics are not that influential […]" originates from the regression analysis. This analysis highlighted GPP as one of the most influential terms in projecting the variability of anoxia in Lake Mendota, but not as influential as physical variables (GPP only explained 15 % of the interannual variance of the Anoxic Factor).

**Referee comment:**
20. Line 420 – Again I agree with you, but other than one variable in seven in the regression, I don't know where this comes from. Need to describe this variables importance.
**Author response:**
As GPP is an ecosystem-scale metric that represents phytoplankton carbon uptake, net aquatic primary production as well as ecosystem respiration it surely highlights the biological control over Anoxic Factor, even if the regression deemed physical variables as more important.

**Referee comment:**
21. Line 425 – Maybe the lack of relations is due to using loading concentrations rather than actual loads. This is what I think the methods say.
**Author response:**
We changed the inflow parameters of total phosphorus and total nitrogen from concentrations to loadings and still the effect of anoxia is low. This is probably due to Lake Mendota's long water residence time of approx. 4 years.

**Referee comment:**
22. Line 433 – Is it loads or concentrations. If it is concentrations, that wouldn't surprise me at all. It is not the annual variations in concentrations that drive things, it is the difference in loads.
**Author response:**
See point above.

**Referee comment:**
23. Line 440 - This could be an important point, maybe there is so much oxygen consumption in the bottom, that it dwarfs any water column consumption. But this disagrees with findings of the other models.
**Author response:**
We discussed the sediment oxygen demand in the main text:

L519-529: The simple deductive model established that the volumetric oxygen sink (i.e. water column oxygen demand) is consistently higher (on average about four times higher) than the sediment oxygen sink. The volumetric sink in lakes has been found to be strongly dependent on the trophic state of the lake, whereas the sediment sink is not (Rippey and McSorley, 2009). Eutrophic lakes tend to have high volume sinks that reach maxima of about 0.23 g m-3 d-1 (Rippey and McSorley, 2009) similar to the average volume sink of 0.16 g m-3 d-1 quantified by the deductive model for Lake Mendota. This finding is confirmed by the works of Conway (1972) who found that the high hypolimnetic oxygen demand of Lake Mendota was driven by algae decomposition, originating from the surface layer. Although eutrophic lakes tend to have a high sediment oxygen demand, the specific values can range from 0.3 g m-2 d-1 (Romero et al., 2004; Steinsberger et al., 2019) to extreme values of 80 g m-2 d-1 (Cross and Summerfelt, 1987), most studies measured or applied a value between 1 to 4 g m-2 d-1 (Mi et al., 2020; Veenstra and Nolen, 1991). The sediment oxygen demand calculated by our deductive model of 0.04 g m-2 d-1 was closer to the average value of approx. 0.08 g m-2 d-1 measured by Rippey and McSorley (2009) on 32 lakes.

In our numerical model the sediment oxygen demand (SOD) is replicating the volumetric and area sink as explained in the "Methods" section. Also, the model SOD is represented over the whole vertical axis (sediment area per volume for each grid cell) instead of a stagnant bottom layer only near lake's bottom. The results of the deductive model did not confirm a very high SOD compared to other eutrophic lakes, see extreme values in Cross and Summerfelt (1987) of up to 80 g per m2 per d.

**Referee comment:**
24. Line 445. The apparent changes caused by the Spiny water flea may be totally confounding any correlations, regressions, and your GLM-AED2 modeling. You may have to stick to one of the periods to really describe the effects of physics vs internal. Or have two different models.
**Author response:**
See our initial response please. We described the Anoxic Factors for the pre-2010 and post-2010 periods in more details and focused our regression analysis only on the pre-2010 period.

**Referee comment:**
25. Line 472. Rather than implementing a different type of dynamic model, maybe better capturing change in productivity and clarity, will help in describing the physics.
**Author response:**
We agree that a better replication of changes in ecosystem-scale metrics like productivity or even water clarity would improve the simulations a lot. Still, water quality models are generally way overparameterized and have problems regarding equifinality. The occurrence of tiny water flea has proven that ecosystem changes will have strong effects on other ecosystem characteristics like anoxia. Therefore, even the best calibrated fixed water quality model will have problems replicating a dynamic ecosystem. Further, our monitoring campaigns do not capture important water quality variables on a high temporal scale, e.g. daily, which generates further uncertainty. Therefore, in our opinion an improvement of the hydrodynamic calculations for example by using a state-of-the-art turbulence closure scheme is the most applicable approach to improve the simulations in the near future.

**Referee comment:**
26. Line 481 – you didn't calibrate the biological parameters, so this should be rewrit-ten.
**Author response:**
We calibrated physical as well as chemical parameters in GLM-AED2 but did not modify the biological parameters of the functional phytoplankton blooms. As these functional variables represent multiple phytoplankton species, a direct calibration would potentially result in an over-calibration of the model for specific time periods, which we tried to avoid. We changed the sentence accordingly to:

L551: Our GLM-AED2 model overestimated spring phytoplankton biomass, which resulted in an overestimation of surface dissolved oxygen concentrations.

**Referee comment:**
27. Line 497 – Rather than thinking the deductive model is biased, maybe it is the only approach capturing the effects of the biology.

**Author response:**

Please see statement above regarding the limitations of the deductive model, and the lines that were revised to better formulate this in the main text.

---

## Author Comment (AC2) · 27 Oct 2020

**Referee general comment:**

This manuscript describes a one-dimensional model (i.e. GLM-AED2) study for Lake Mendota which analyzed its long-term changes of anoxia and the driving factors. As a major result, the model showed good performance in reproducing oxygen dynamics, especially the low oxygen concentration in the hypolimnion,in the lake and based on the statistical analysis, it suggested that the physical structure (e.g. Schmidt Stability, onset of stratification, water temperature in the hypolimnion) had a big influence on the spatial and temporal development of anoxia. This is an interesting and important study, which could be considered for publication after a minor revision. Although there are quite a few studies analyzing hypolimnetic anoxia for inland waters, most of them draw their conclusion based on the short-term measurements and there is still a need to comprehensively illustrate this phenomenon and mechanisms behind its formation based on long-term database. Based on this prospective, this research fills in a research gap. In my opinion, this paper is well organized and its content, especially the discussion part will improve our understanding about anoxia and its future development under climate warming. Detailed comments are shown below.

**Referee comment:**

2.1 Study Site: It is better to show a topographic map of this lake, as well as the location for the water quality measurements.

**Author response:**

Thank you for this suggestion. We have added a new figure to the manuscript that shows the location and landuse overview of Lake Mendota, as well as the location of the measurement stations.

[Figure]

Figure 1 Location and overview map of Lake Mendota, Wisconsin, which is located in the Yahara River catchment in southern Wisconsin, USA. USGS gage stations for the PIHM-Lake model and the location of the Lake Mendota monitoring buoy are placed in the map. Land cover was obtained from the US National Land Cover database.

**Referee comment:**

L 115: 1.How you calibrated the hydrological model?

**Author response:**

The hydrological PIHM-Lake model was calibrated to measured stream inflows to the lake and outflow discharges from the lake to the catchments. The model was calibrated by using the observations from 2009 to 2011 and validated by using the measurements from 2012 to 2014, within which all stream flow observations are available. To state this clearer in the main text, we slightly modified this sentence in the manuscript:

> L125: The PIHM-Lake simulation covers a 37-year period (from 1979 to 2015), and its parameters were calibrated and validated with *in-situ* measured stream inflow and lake outflow discharges from the US Geological Survey.

We also attached the following figure to this reply here, which shows the fit between observed discharges of three Lake Mendota inflows (Pheasant branch, Six Mile, Yahara) and the outflow from Lake Mendota to the simulated discharges by PIHM-Lake (calibrated).

[Figure]

**Referee comment:**

2.From I know for the historical simulation, the inflow discharge is always drawn from the real measurements, instead of hydrological models. Do you have the measured inflow discharge for Lake Mendota?

**Author response:**
Yes, we have used measured inflow discharges for Lake Mendota at 4 inflows gages, see Fig. 1, but these monitoring stations only present an incomplete water balance as all groundwater inflow and surface overland flow to Lake Mendota are not observed, which could also contribute to the water balance of Lake Mendota. Therefore, we additionally used a calibrated hydrological PIHM-Lake model (using monitored flow data and lake surface water level fluctuations) to create two general inflows terms that close the overall lake water balance. To clarify this, we have added a sentence to the main text:

> L127: The application of the PIHM-Lake model for quantifying the lake inflows helped closing the water balance of Lake Mendota as groundwater inflow and surface overland flow were not measured, and the model simulations provided these inflows.

**Referee comment:**
L 125: How many types of nutrients were included here as the inflow boundary conditions? It is better clarify it here.

**Author response:**
Thank you for pointing this out. We included a sentence in the main text:

> L133-136: We included the following nutrients in the inflow boundary conditions soluble reactive phosphate, adsorbed soluble reactive phosphate, dissolved organic phosphorus, particulate organic phosphorus, dissolved organic nitrogen, ammonia, nitrate, refractory dissolved organic carbon, dissolved inorganic carbon, and reactive silica.

**Referee comment:**
L133: I am not sure whether it is appropriate to define the inflow loading as the mean values from the water column. It means that there is no seasonal changes of DIC and silica, which is unrealistic. Could you explain why you set the inflow DIC and silica in this way?

**Author response:**
After a long internal discussion, we set DIC and silica to an average value as these variables are not part of the routine measurement program. As the average in-lake value is quite high, we did not expect any sensitivity of these values on the model results. Further, in-lake measurements have shown that the average concentration in the lake does not fluctuate much.

**Referee comment:**
2.3 Modelling Framework: Just a recommendation, it may be better to combine 2.3 to 2.7 into one part, since all of such content be longs to the model description.

**Author response:**
In accordance with the reviewer's suggestion, we changed the levels of sectioning of these paragraphs, e.g. "Deductive Model", "GLM-AED2", "Post-Processing of GLM-AED2 Output" and "Regression Model" are now all sub-paragraphs of "2.3 Modeling Framework".

**Referee comment:**
L 198: For water temperature simulation, I supposed the most important parameters should be wind factor and light extinction coefficient. How you defined these two in the model?

**Author response:**
For identification of calibration parameters, we used the Morris Sensitivity method, which declared the short-wave solar radiation factor, the long-wave radiation factor, the bulk aerodynamic sensible heat transfer coefficient, and the sediment temperatures as the most sensitive model parameters. Therefore, we did not calibrate the wind factor and left it a 1.0, e.g., we used the measured wind data from a close airport. The light extinction coefficient was set to a low water background value of 0.1, because the value was dynamically changing in the water quality model AED2, which backfed any changes in light extinction due to abundance of dissolved substances to the hydrodynamic model. We checked the dynamic modeled light extinction values with measured Secchi depth data, and the seasonal dynamics were replicated by the model.

**Referee comment:**

L 293: How you calculated GPP? It is better to clarify it here.
**Author response:**
GPP (in mmol C per m3 per d) was internally calculated by the AED2 model as the daily total carbon uptake of all functional phytoplankton groups. We clarified this in the main text:

> L300-308: Here, GPP represents the sum of all functional phytoplankton group's photosynthesis rates parameterized as the total carbon uptake:
> $$f_{uptake}^{PHY_C} = R_{growth}^{PHY}(1 - k_{pr}^{PHY}) \, \phi_{temp}^{PHY}(T) \, \phi_{stress}^{PHY}(X) \, min\{\phi_{light}^{PHY}(I) \, \phi_N^{PHY}(NO_3, NH_4 PHY_N) \, \phi_P^{PHY}(PO_4, PHY_P) \, \phi_{Si}^{PHY}(Rsi)\}[PHY]$$
> (7)
> where the carbon uptake $f_{uptake}^{PHY_C}$ of an individual group *PHY* depends on the growth rate $R_{growth}^{PHY}$, the photorespiratory loss $(1 - k_{pr}^{PHY})$, temperature scaling $\phi_{temp}^{PHY}(T)$, metabolic stress $\phi_{stress}^{PHY}(X)$, and a minimum function taking into account limitations by light $\phi_{light}^{PHY}(I)$, nitrogen $\phi_N^{PHY}(NO_3, NH_4 PHY_N)$, phosphorus $\phi_P^{PHY}(PO_4, PHY_P)$ and silica $\phi_{Si}^{PHY}(Rsi)\}$ (Hipsey et al., 2017; adapted from Hipsey and Hamilton, 2008). As the GPP is the main model output variable for phytoplankton dynamics, it scales directly with biomass and Chl-a concentrations.

**Referee comment:**
L 333: There existed some negative values for Birgean Work in Figure 5, what is the reason for that?
**Author response:**
As the Birgean Work is

$$B = \int_0^{z_m} A_z(1 - \rho_z)z dz$$

a negative value can occur when a dominant part of the water column has water densities that are above 1,000 kg per m3. Hypothetically speaking, a negative Birgean value would mean that no energy is needed (or negative energy would be needed) to achieve the current stratification from a completely mixed state, which means that the current state is probably also completely mixed. We decided against discussing this in the manuscript as we focused on oxygen dynamics over time.

**Referee comment:**
L 371: In Figure 9B, why was the simulated AF represented by dots, instead of box plots as the measured one?
**Author response:**
The simulated AF is represented by dots as we calculated it from the GLM-AED2 output and were therefore able to quantify it using daily data. On the other hand, the observed data were only available every two weeks, therefore we used different interpolation techniques to get daily data. These uncertainties were captured in a box-plot.

---

## Author Response (AR2)

**General Response:**
Thank you for the revisions, which all have substantially improved our manuscript since the first submission. We felt the reviews were particularly thoughtful regarding the uses of alternative models to better understand and predict lake Anoxic Factor. The questions and criticisms raised by the reviewers led to a more careful and thorough description of our deductive modeling approach, which makes the results easier to understand and interpret. In addition, the reviewer points out a modeling situation that is easily overlooked – what we learn from the noise can be just as interesting as what we learn from the signal. The residual error from our predictions of Anoxic Factor shows an upward shift in anoxic factor in 2010, indicating either an unobserved change in drivers or an important process missing from the model at that time. While it is easy to be critical of the model for missing the shift, it perhaps is more important to use the missed shift to reflect on our knowledge of the ecosystem and to think of ways to attack this newly found problem. We are particularly grateful to the reviewers for encouraging us to pursue this line of reasoning.

The manuscript has improved substantially as an outcome of the review process. We thank the reviewers for their time and their valuable critiques. We hope the latest draft meets your expectations. Most of our response here is focused on the use of Chl-a and the description of the deductive model. Both of these have limitations, but we believe we have thoroughly addressed the reviewer's concerns. In the few cases where we do not incorporate their suggestions, we provide the details behind our decision making.

**Referee major comment:**
I think the authors should also try to use regression as an independent approach to further validate the results of the dynamic model. Normally I would prefer that the regression model ONLY use non-modeled information to make any conclusions; however, I realize that given the frequency of data collection this may be very difficult. Therefore, given that GLM-AED model simulates the physics quite well, I think the authors should run one more regression. That regression would use actual summer average Chl-a instead of GPP in the regression. This would show whether the lake actually behaves like GLM-AED says it does.

**Author response:**
We tested this approach, but unfortunately it did not yield the results we thought it might. Below we plot the relationship between the sum of measured vertical Chl-a concentration (winter to spring) against observed average Anoxic Factors (Fig R1, similar to the regression of modeled winter-spring GPP to summer Anoxic Factor). The relationship is weak. A linear model AF ~ Chl-a with the limited data from 2008-2016, returns $R^2 = 0.03$ with p = 0.29.

We believe three factors are at play (1) winter and under-ice measurements are rare, therefore we are potentially underestimating these values, (2) Chl-a, as a state variable, is only a rough proxy for the process, GPP, and (3) the observed Chl-a data on Lake Mendota for the period 2002-2007 is suspect due to, "an uncorrectable bias" of Chl-a data due to a change in instruments:
(see abstract information here https://doi.org/10.6073/pasta/f28e278afc34f1b7bd4f3cdc02b733a2).

[Figure]

*Figure R1: Scaled observed average AF against scaled observed vertical sums of Chl-a prior to summer.*

Even though Chl-a is not a suitable predictor for AF, we agree that the influence of biology on AF merits emphasis in the manuscript. In the manuscript, the GPP prior to summer (winter to spring) was the main influential factor for summer anoxia (as stated e.g. L479: "Gross primary production (GPP) in the epilimnion prior to summer stratification is a secondary, but still important, predictor of anoxia." and in Table 1). To make this clearer, we revised all mentions of GPP as important predictor in text to "GPP prior to summer" in the manuscript. The settling of POC into deeper water layers prior to stratification is therefore the main process affecting anoxia, as stated e.g. at L480: "GPP fuels the sinking of particulate organic carbon (POC) into deeper layers before the establishment of a thermocline. In the hypolimnion, POC is readily decomposed into DOC and mineralized by bacteria in the numerical model, and reflects the dissolved oxygen volume sink."

**Referee major comment:**
My other concern is in the presentation of the results of the deductive model. First, describe how J(z) is actually computed, describe what we are seeing in Figure 3, use one year for example, and then describe what the volumetric part of this model really means. My first impression was that this model was giving very different results than the other two models. But I can see the volumetric part of the model may also represent variability in productivity and changes in stratification (although this is not described in the Discussion). I think my major confusion here is with the description of the model and Figure 3. Personally, I don't like any approach where I am interpreting the slope and intercept of noisy data. As data get noisy the slope goes closer to 0 and thus changes the overall interpretation.

**Author response:**
We revised the text in "2.3.1 Deductive Model" and added more information to it:

> L178: Using temporal and spatial linearly interpolated observed dissolved oxygen data, we applied the simple deductive oxygen depletion model according to Livingstone and Imboden (1996) in which the oxygen depletion rate *J(z)* at depth *z* is conceptualized as
> $$J(z) = J_v(z) + J_A(z)\alpha(z), \tag{1}$$
> Where the intercept $J_V$ is the volume sink (mass per volume per time) representing organic matter mineralization processes, e.g. microbial respiration in the water column, the gradient $J_A$ is the area sink (mass per area per time) representing sediment oxygen demand, and $\alpha$ is a function for the $\alpha(z)$ratio of sediment area to water volume over the depth *z* (Bossard and Gächter, 1981; Livingstone and Imboden, 1996):

$$\alpha(z) = -\frac{1}{A(z)}\frac{dA(z)}{dz}. \tag{2}$$

We used observed dissolved oxygen data from 1992 to 2015 (measured biweekly after ice offset) to calculate the specific oxygen depletion $J(z)$ over depth for each year individually from the concentration, $[DO]_{spring}$, at the date of spring mixing offset, $t_{spring}$, to the date, $t_{2mgL}$, when oxygen concentrations, $[DO]_{2mgL}$, were below 2 mg L-1 (criterium for hypoxia):

$$J(z) = \frac{[DO]_{spring} - [DO]_{2mgL}}{t_{spring} - t_{2mgL}}. \tag{3}$$

Only dissolved oxygen data below a depth of 15 m were used. The derivatives of area to depth were approximated by using forward and backward differencing. The terms $J_V$ and $J_A$ were assumed to be constant for every year (assuming the hypolimnion to be homothermic) and were determined by using weighted linear regression.

We further revised "3.1 Oxygen Depletion Rates" to make it clearer that the volumetric and areal sinks are represented by the intercept and gradient, respectively:

L343: The derived annual oxygen depletion rates by the deductive model confirmed Lake Mendota's hypolimnetic anoxia as primarily driven by mineralization of organic matter. Observed oxygen depletion rates, J(z), and against area-volume ratios, $\alpha(z)$, were positively correlated for all years except 1993, 1997 and 2007 (Figure 3). For years with a positive relationship, the average intercept representing the volumetric sink JV as was 0.16 g m-3 d-1 and the average gradient representing the areal sink JA with was 0.04 g m-2 d-1 (adjusted R2 = 0.13, p < 0.001). Lake Mendota's hypolimnetic oxygen depletion was mainly driven by water column respiration mineralization processes over sediment oxygen demand. The annual volumetric depletions rate followed a normal distribution with an increase in the volumetric sink in recent years. The areal depletion rate distribution was positively skewed. An inspection of the residuals from the model fits indicates that the linear regression model may not be appropriate for some years, especially for values of the sediment to area volume ratio $\alpha(z)$ near 0.5 m2 m-3.

Regarding the description of net ecosystem production terms or physical drivers: First, we recognize that the observed data is influenced by physical as well as biogeochemical drivers. Further, the deductive model according to Livingstone and Imboden (1996) is based on the radon and phosphorus model from the Imboden and Emerson (1978) paper, in which all sink terms are described by the term $J$. But, as in the deductive oxygen model, production and vertical transport of dissolved oxygen are neglected, the volumetric sink (or the intercept in the linear regression), $J_v$, does mathematically only represent ecosystem respiration/mineralization in the water column (see also Charlton 1980, or Mathias and Barica 1980). The deductive model therefore can only describe negative aquatic ecosystem production processes, in which ecosystem respiration is higher than gross primary production. Vertical transport by i.e. turbulent eddy diffusion is neglected, therefore the volumetric processes do not represent the physics. Of course, the field data is influenced by stratification onset and the limitation of vertical fluxes, but the simple linear regression assumes that any changes in vertical fluxes are neglectable. Therefore, we decided to use the results from the deductive model as support for our sediment oxygen demand value in the process-based model, GLM-AED2. Additionally, we decided to discuss the results of the deductive model in "4.3. Biological Control over Anoxic Factor", as it can only quantify the biochemical oxygen sink terms from observed data.

Imboden, D.M., and Emerson, S. 1978. Natural radon and phosphorus as limnologic tracers: horizontal and vertical eddy diffusion in Greifensee. Limnol. Oceanogr. 23: 77–90.
Charlton, M.N. 1980. Hypolimnion oxygen consumption in lakes: discussion of productivity and morphometry effects. Can. J. Fish. Aquat. Sci. 37: 1531–1539.
Mathias, J.A., and Barica, J. 1980. Factors controlling oxygen depletion in ice-covered lakes. Can. J. Fish. Aquat. Sci. 37: 185–194.

**Minor Comments**

**Referee comment:**
1. Line- 21. Remove the word "evolutionary".
**Author response:**
Technically, the CMA-ES algorithm belongs to the group of evolutionary optimization algorithms that mimic biological evolution to find a global optimum for a given function. To avoid confusion, we agree that eliminating all mentions of "evolutionary" in the manuscript is warranted.

**Referee comment:**
2. Line 25 and later. I think the real strength in a regression model is to provide independent information that the dynamic model is simulating reality. See suggestion above.
**Author response:**
Please see our reply to the referee's first major comment above, in which we regress observational data per the referee's suggestion. Based on the limitations discussed in our first reply (data scarcity, potential bias), we used linear regression on modeled data as previously done in Snortheim (2017), Ward (2020) and Weng (2020). Here, all assumptions of the manuscript were done in model space and we recognize the constraints of the model, although we aimed to minimize potential bias by calibrating it to the best of our knowledge and data availability.

**Referee comment:**
3. Line 30. Make it read "a measured step upward".
**Author response:**
Agreed, we revised the text accordingly:

> L31: A measured step change upward in summer anoxia in 2010 was unexplained by the GLM-AED2 model.

**Referee comment:**
4. Line 48. There is a decadal shift in anoxia in Lake Mendota, and this should be brought into the final discussion a little better. This may be a major difference in what Snortheim described (line 60).
**Author response:**
Agreed. We revised the text in "4.3 Biological Control over Anoxic Factor" to discuss Snortheim et al. more and highlight the connection to spiny water flea invasion:

> L510: The model replicated the maximum anoxia event in 1998 but struggled to replicate the minimum in 2002. The discrepancies of 5-10 days between the simulated and observed range of the Anoxic Factor beginning in 2010 are related to an increased spatial as well as temporal extent of summer anoxia (Supplement Figure A10), which was not captured by the model. A similar increase in observed Anoxic Factors starting in 2010 was also visualized in the study by Snortheim et al. (2017), but possible causes were not discussed. This The increased spatial as well as temporal extent of summer anoxia was highlighted by the statistical analysis of the pre-2010 (1992-2009) and post-2010 (2010-2015) Anoxic Factors. Prior to 2010, there were no significant differences between observed and modeled distributions (p=0.13); whereas, after 2010, the observed distribution was significantly higher than the modeled distribution (p=0.032) (Supplement Figure A9). Similarly, the pre-2010 observed Anoxic Factors were significantly different than the post-2010 observed Anoxic Factors (p=0.0049). For simplicity and due to limitations in Lake Mendota monitoring data post-2010, we focused the regression analysis of the Anoxic Factor in this study only on the pre-2010 period. The detection of this decadal shift in summer anoxia post-2010 highlights a hidden biological process that was not considered in the process-based model and may be due to an ecosystem shift in Lake Mendota that began in 2009, when the invasive spiny water flea (Bythothrephes longimanus) was detected in surprisingly high densities in the lake (Walsh et

al., 2016b, 2018). Spiny water flea effectively became the dominant Daphnia grazer, causing historically low Daphnia biomass in 2010, 2014 and 2015 (Walsh et al., 2016a) and reducing water clarity. The spiny water flea may have increased organic matter supply to the hypolimnion by grazing down certain phytoplankton. Mendota's Daphnia population historically consisted of Daphnia pulicaria and the smaller-bodied Daphnia galeata mendotae, who compete differently with spiny water flea. D. mendotae biomass increased in spring after the spiny water flea invasion (Walsh et al., 2017), grazing on phytoplankton and probably accelerating organic matter mineralization before stratification onset. This could be one potential cause that contributed to the increase in hypolimnetic oxygen depletion after 2010. Our GLM-AED2 model could not replicate this food web change, and subsequent shift in anoxia dynamics, due to limitations of the numerical model, i.e., GLM-AED2 had constant ecological parameters over the entire modeling period and did not have zooplankton dynamics instantiated. We envision future monitoring and modeling studies of Lake Mendota that focus entirely on ecosystem shifts associated with the invasion of spiny water flea in 2009 and the exponential growth of zebra mussels from 2015-2018 (Spear, 2020).

We added two sentences regarding the decadal change to "5 Conclusions":

> L625: Further, our modelling framework detected a decadal shift in the Anoxic Factor starting in 2010, which was not replicated by our process-based model and therefore probably not driven by physical or chemical drivers, but related to an ecosystem shift caused by the invasive Bythothrephes longimanus.

**Referee comment:**
5. Line 83. Need to be careful here. Just because the model has high frequency output, it may not represent what is really happening in the lake. Empirically evaluating results of dynamic models may describe the mathematical equations in the model, but not how this particular lake actually works.
**Author response:**
We agree, but in this study we decided to evaluate emergent ecosystem characteristics by working in model space and calibrating the process-based model to best of our knowledge and data. We revised that line to:

> L83: Results from deterministic lake models can be analysed using statistical models to derive general relationships of cause and effect in the model space.

**Referee comment:**
6. Line 92. You state that you are going to use data driven empirical models to evaluate observed data, that is really a good idea, and I think you need to do this more. Maybe by using Chl-a, you can get to this.
**Author response:**
Please see our reply to the referee's first major comment above.

**Referee comment:**
7. Line 96. I think you should add something about the decadal changes in Lake Mendota. Also state this in your Conclusions. Because the models don't capture it, it suggests something outside of the physics and chemistry is driving it. This is a strength of the overall approach.
**Author response:**
Agreed, we added a sentence regarding the decadal change to "5 Conclusions":

> L625: Further, our modelling framework detected a decadal shift in the Anoxic Factor starting in 2010, which was not driven by physical or chemical drivers, but probably related to an ecosystem shift caused by the invasive Bythothrephes longimanus.

**Referee comment:**

8. Line 117. I still have a problem with PIHM-Lake never really being presented in this paper or published elsewhere.

**Author response:**

Unfortunately, a related publication about PIHM-Lake is still undergoing the review process. Therefore, we added multiple paragraphs to the supplement that explain PIHM-Lake in more details.

Supplement text:

**PIHM-Lake description**

PIHM-Lake is built upon a physically-based spatially distributed hydrologic model—PIHM (Penn State Integrated Hydrologic Model) (Qu and Duffy, 2007)—with the capability of simulating surface, subsurface, and channel water exchange between a catchment and a lake, as well as the water level change of the lake. As illustrated in Supplement Figure A11, PIHM-Lake model uses a finite volume numerical scheme and unstructured triangular mesh to represent the domain. It tracks the changes of surface and subsurface water storage on a 3D catchment and 1D lake as a function of precipitation, evapotranspiration, recharge, surface and groundwater flow, channelized flow, and snow melt. The spatial variation of overland flow and groundwater flow between the catchment and the lake is characterized by the water flows through the edges of each triangular mesh. Specifically, based on the conservation of mass of water, the generic form of the governing equations for PIHM-Lake is

$$\frac{dS_{canopy}}{dt} = vFrac * (1 - sFrac) * P - E_c$$

$$\frac{dS_{snow}}{dt} = sFrac * P - SM$$

$$\frac{\partial S_{surf}}{\partial t} = TF - \nabla q_{sw} - I - E_s$$

$$\frac{dS_{unsat}}{dt} = I - R - E_g - E_{gt}$$

$$\frac{\partial S_{sat}}{\partial t} = -\nabla q_{gw} + R - E_{sat} - E_{tsat}$$

where $\frac{dS_{canopy}}{dt}$ = the time rate of change of the canopy water storage, $S_{canopy}$ (m), due to canopy evaporation $E_c$ (m/day) and canopy interception $vFrac * (1 - sFrac) * P$ (m/day). $vFrac$ and $sFrac$ are the vegetation fraction and snow fraction, respectively. $P$ = precipitation (m/day). $\frac{dS_{snow}}{dt}$ = the time rate of change of snow storage $S_{snow}$ (m) due to $sFrac * P$:snow formation from precipitation when temperature is below 0 ºC (m/day) and *SM*, snow melt (m/day), which is a function of degree-day factor of ice and snow melt. $\frac{\partial S_{surf}}{\partial t}$ =the time rate of change of surface water storage, $S_{surf}$ (m), due to *TF*= throughfall (m/day), $\nabla q_{sw}$= net overland flow (m/day), *I*: infiltration (m/day), and $E_s$: surface water evaporation (m/day). $\nabla q_{sw}$ is modeled by the diffusion wave approximation of St. Venant's equation assuming shallow surface water depth and negligible influence of inertia force on overland flow, which is equivalent to Manning's equation. The estimation of infiltration rate is a function of the gradient of the surface and subsurface hydraulic head. $\frac{dS_{unsat}}{dt}$ represents the time rate of change of unsaturated water storage (m) due to *I*: infiltration (m/day), *R*: recharge (m/day), $E_g$: soil evaporation (m/day), and $E_{gt}$: transpiration (m/day). The recharge is calculated using Richard's equation assuming a vertical exchange of water across a moving water table interface. $\frac{\partial S_{sat}}{\partial t}$= the time rate of change of $S_{sat}$: the saturated water storage (m). $\nabla q_{gw}$ = net groundwater lateral movement between adjacent cells (m/day) is

represented by the Darcy-type flow proportional to groundwater gradient. $E_c$, $E_s$, $E_g$ and $E_{sat}$ are the evaporation (m/day) from the vegetation canopy, surface water, unsaturated and saturated soil zone, respectively. The potential evaporation rate is estimated by the Penman equation. The transpiration (m/day) is described by $E_{gt}$ or $E_{tsat}$, depending upon the vegetation coverage, the rooting depth and the groundwater table. If the groundwater table is higher than rooting depth, plants uptake water from the saturated zone, and $E_{tsat}$ applies. Otherwise, water uptake occurs at the unsaturated soil zone, and $E_{gt}$ applies.

For the hydrodynamics of the 1-D lake, we consider a two-layer system: a surface water layer and an aquifer layer. Surface water flow between the catchment boundary cells directly affects the water storage of surface water layer. Meanwhile, subsurface water flows through the aquifer layer and indirectly contributes to surface water through negative recharge. Likewise, based on the conservation of mass of water, the governing equation for the 1D lake component is

$$\begin{cases} \dfrac{\partial S_{lake\_surf}}{\partial t} = P - E_s - R + q_{sw} \\ \dfrac{\partial S_{lake\_gw}}{\partial t} = R + q_{gw} \end{cases}$$

where $\dfrac{\partial S_{lake\_surf}}{\partial t}$ = the time rate of change of lake surface water. $\dfrac{\partial S_{lake\_gw}}{\partial t}$ = the time rate of change of water storage in lake bottom aquifer. $P$ =precipitation (m/day); $E_s$ = surface water evaporation (m/day); $R$ = recharge (m/day). A positive value of R indicates downward lake surface water, while a negative value indicates an upward groundwater recharge to surface water; $q_{sw}$ and $q_{gw}$ are surface and groundwater flow through the edges of the lake boundary, respectively.

Details of the model processes and code is referred to the model repository: https://github.com/hydro-geomorph-zhang/PIHM-Lake.

[Figure]

*Figure A11 Conceptual framework of PIHM-Lake.*

**Referee comment:**
9. Line 134. You have all kinds of nutrient components. I think you need to describe the assumptions you made to not only go from TP and TN to all of them. Why would you double the only thing that you actually measured (Line 137)?
**Author response:**

We used measured concentrations without any assumptions for the inflow loading regressions of all water quality variables except phosphorus (as described in the manuscript), refractory organic matter, dissolved inorganic carbon and silica. For these variables, inflow concentrations were not available, except for phosphorus, and so we used constant value for the loadings similar to the long-term averages measured in the lake. For phosphorus, we doubled the measured TP concentration to account for adsorbed phosphate, which is easily underestimated in manual sampling programs. Most studies underestimate these loads which become increasingly more important due to extreme storm events (see Carpenter et al. 2018).

**Referee comment:**
10. Line 139. Didn't Lathrop present measured/actual loading to Lake Mendota in several papers? Seems funny that those estimates are still not mentioned.
**Author response:**
Thank you for mentioning this point. We had the calculations by Lathrop on hand here but decided to focus on the estimates by Bennett as well as Kara. We added estimates of annual TP loads by Lathrop and Carpenter to the text:

> L141: Our average annual TP load (without adsorbed phosphate) was about 25.3 t and ranged between 2.7 to 73.1 t (1979-2015), which is similar to previous annual TP load estimates between of 15 to 67 t (Kara et al., 2012) and 10 to 80 t (Lathrop and Carpenter, 2014).

**Referee comment:**
11. Line 150. I think a reference is needed for the 1992-1994 data.
**Author response:**
The additional oxygen data were sampled during the graduate work of Patricia Soranno, but were not part of any publication to the best of our knowledge. We acknowledged her work and data, and added her Doctoral Thesis from 1995 as reference here:

> L149: The dissolved oxygen data set was complemented with historical measured dissolved oxygen data from 1992 to 1994 (Soranno, 1995).

We hope that in the near future we can add her data to the NTL-LTER data repository.

**Referee comment:**
12. Line 180. Describe how J(z) is actually computed and how Jv and Ja are estimated from the slope and intercept of the relation between J(z) and alpha. So does each point in Figure 3 represent a different depth?
**Author response:**
We revised the text in "2.3.1 Deductive Model":

> L178: Using temporal and spatial linearly interpolated observed dissolved oxygen data, we applied the simple deductive oxygen depletion model according to Livingstone and Imboden (1996) in which the oxygen depletion rate $J(z)$ at depth $z$ is conceptualized as
> $$J(z) = J_v(z) + J_A(z)\alpha(z), \qquad (1)$$
> Where the intercept $J_V$ is the volume sink (mass per volume per time) representing organic matter mineralization processes, e.g. microbial respiration in the water column, the gradient $J_A$ is the area sink (mass per area per time) representing sediment oxygen demand, and $\alpha$ is a function for the ratio of sediment area to water volume over the depth $z$ (Bossard and Gächter, 1981; Livingstone and Imboden, 1996):
> $$\alpha(z) = -\frac{1}{A(z)}\frac{dA(z)}{dz}. \qquad (2)$$
> We used observed dissolved oxygen data from 1992 to 2015 (measured biweekly after ice offset) to calculate the specific oxygen depletion $J(z)$ over depth for each year individually from the concentration, $[DO]_{spring}$, at the date of spring mixing offset, $t_{spring}$, to the date, $t_{2mgL}$, when oxygen concentrations, $[DO]_{2mgL}$, were below 2 mg L-1 (criterium for hypoxia):

$$J(z) = \frac{[DO]_{spring} - [DO]_{2mgL}}{t_{spring} - t_{2mgL}}. \tag{3}$$

Only dissolved oxygen data below a depth of 15 m were used. The derivatives of area to depth were approximated by using forward and backward differencing. The terms $J_V$ and $J_A$ were assumed to be constant for every year (assuming the hypolimnion to be homothermic) and were determined by using weighted linear regression.

**Referee comment:**
13. Line 225. Remove the word evolutionary.
**Author response:**
Agreed.

**Referee comment:**
14. Line 273. Is there any way to describe how you really combined simulated DO and measured DO to get the AF? Was the real data always used and then interpolated with simulated data?
**Author response:**
Yes, real data is always used and interpolated to determine the temporal and spatial extent of summer anoxia.
Method: Our observed data was bi-weekly during the ice-free period, therefore we needed to apply interpolation techniques to approximate DO values on a daily grid with a higher vertical resolution (as this matters for the determination of AF). Therefore, we used three different interpolation techniques, namely linear, constant and spline. In the final Fig. 10, modeled data were visualized as point values, whereas observed Anoxic Factors needed to be visualized as box-plots. We revised the text accordingly:

> L276: Observed Anoxic Factors were calculated by temporally and spatially interpolating bi-weekly monitored field data, using an ensemble of approaches (linear, constant and spline interpolation between neighboring data points). We quantified the seasonal Anoxic Factor only for the summer season, respectively for the modeled and observed data. We then compared the modeled Anoxic Factor (quantified by using modeled daily dissolved oxygen data profiles) against a set of observed Anoxic Factors (here, the bi-weekly data were temporally and spatially interpolated to get daily estimates over a finer vertical resolution) that were obtained by the application of three interpolation techniques.

**Referee comment:**
15. Line 278 and 324 and 421. Can you make this into two regression models? One the way you did it and one with Chl-a?
**Author response:**
Please see our reply to the referee's first major comment above.

**Referee comment:**
16. Line 317. Something to consider for the future. In the regression model add a variable to represent the change in time: 0 for the first half and 1 for the second half. Then you can see if the change was significant.
**Author response:**
Thank you very much for this very helpful suggestion!

**Referee comment:**
17. Line 337. See comments above about explaining Figure 3. It would help to state that 0.16 is the average intercept and 0.04 is the average slope from all of the figures. Remove the word respiration, this is what gets confusing. By removing the word respiration, then physics is still in this part.

**Author response:**

Thank you. We changed the text accordingly:

> L344: Observed oxygen depletion rates, J(z), against area-volume ratios, $\alpha(z)$, were positively correlated for all years except 1993, 1997 and 2007 (Figure 3). For years with a positive relationship, the average intercept representing the volumetric sink $J_V$ as was 0.16 g m-3 d-1 and the average gradient representing the areal sink $J_A$ with was 0.04 g m-2 d-1 (adjusted R2 = 0.13, p < 0.001). Lake Mendota's hypolimnetic oxygen depletion was mainly driven by water column respiration mineralization processes over sediment oxygen demand.

But vertical transport by i.e. turbulent eddy diffusion is neglected, therefore the volumetric processes do not represent the physics (see response to general comment at the beginning). Although we do recognize that the field data is influenced strongly by physical processes, the regression model mathematically does not incorporate these considerations.

**Referee comment:**

18. Line 343. Why would you add both pieces to get an estimate of SOD, shouldn't you only use the 0.04?

**Author response:**

In the GLM-AED2 model, the DO equation is mainly based on atmospheric exchange plus the sediment oxygen demand, which represents, in a conceptual way, the total oxygen sink over the water column. As bacterial mineralization in AED2 is based on temperature- and oxygen-dependence, we decided – conceptually – to use the SOD value of the model as the sink for oxygen, hence the main model compartment for the oxygen depletion rate. Therefore, we applied the total depletion rate, quantified by the deductive model, as the model's sediment oxygen demand, as "internal fluxes of organic carbon from the sediment back into the water column would drive additional oxygen depletion." (L355 in the main manuscript).

**Referee comment:**

19. Line 405. Should reference Table 2. I don't think your RMSEs are similar to that referenced – they are bit higher.

**Author response:**

Thank you for pointing this out. We revised the text accordingly:

> L413: Dissolved oxygen dynamics, including the spatial extent of oxygen depletion in the water column, and the timing of summer anoxia periods, were replicated by the GLM-AED model (Figure 9A-B, Table 2); although the model overestimated spring and summer time surface oxygen concentrations due to a higher net ecosystem production. The depth-averaged fit criteria of dissolved oxygen concentrations were similar but slightly higher to a recent study from Farrell et al. (2020) […].

**Referee comment:**

20. Line 416. Rather than saying the AF has no significant differences, use the model not capturing things after 2010 as a strength and that there are decadal changes occurring in the lake.

**Author response:**

Thank you, we highlighted the detection of the decadal shift in anoxia in the discussion and conclusions (see other replies). Here in "3.4 Oxygen Dynamics" we highlight that the model's simulated Anoxic Factors were similar to the ones observed pre-2010, but significantly different in post-2010. We revised the text:

> L426: A subsequent Wilcoxon signed-rank test highlighted that the observed average and modelled Anoxic Factors from the pre-2010 period showed no significant differences between the two distributions, suggesting they belong to the same population (p-value = 0.13, Supplement Figure A9 A), whereas the distributions of observed mean Anoxic Factors and modeled ones after 2010 were significantly different (p-value = 0.032, Supplement Figure A9 B), highlighting a potential decadal shift in oxygen depletion patterns.. On the contrary, the modeled Anoxic Factor

distributions of the pre- and post-2010 period were not significantly different (p-value = 0.49, Supplement Figure A9 C), whereas the distributions of the observed Anoxic Factors were significantly different (p-value = 0.0049, Supplement Figure A9 D).

Additionally, we revised Supplement Fig. A9 to also highlight the differences between pre- and post-2010 modeled and observed Anoxic Factors, respectively:

[Figure]

Fig. A9:

*Figure A9 Box-whisker plots of (a) observed to modeled Anoxic Factors for the pre-2010 period 1992-2009, (b) observed to modeled Anoxic Factors for the post-2010 period 2010-2015, (c) pre- to post-2010 modeled Anoxic Factors, and (d) pre- to post-2010 observed Anoxic Factors.*

**Referee comment:**

21. Line 432. Rather than ignoring the results of the deductive model, add a line here about it representing all volumetric processes including the physics.

**Author response:**

The deductive model according to Livingstone and Imboden (1996) is based on the radon and phosphorus model from the Imboden and Emerson (1978) paper, in which all sink terms are described by the term $J$. As in the oxygen model, production and vertical transport of dissolved oxygen are neglected, the volumetric sink (or the intercept in the linear regression), $J_v$, does only represent ecosystem respiration/mineralization in the water column (see also Charlton 1980, or Mathias and Barica 1980). Vertical transport by i.e. turbulent eddy diffusion is neglected, therefore the volumetric processes do not represent the physics. Still, we do recognize that physical changes in the system influence the relationships and measured concentrations of the observed data, and therefore i.e. stratification onset, plays an important role, although any physical transport is mathematically neglected by the simple linear regression approach. We use the

results from the deductive model as support for our sediment oxygen demand value in the process-based model, GLM-AED2. Additionally, we discuss the results of the deductive model in "4.3. Biological Control over Anoxic Factor", as it can only quantify the biochemical oxygen sink terms from observed data.

Imboden, D.M., and Emerson, S. 1978. Natural radon and phosphorus as limnologic tracers: horizontal and vertical eddy diffusion in Greifensee. Limnol. Oceanogr. 23: 77–90.
Charlton, M.N. 1980. Hypolimnion oxygen consumption in lakes: discussion of productivity and morphometry effects. Can. J. Fish. Aquat. Sci. 37: 1531–1539.
Mathias, J.A., and Barica, J. 1980. Factors controlling oxygen depletion in ice-covered lakes. Can. J. Fish. Aquat. Sci. 37: 185–194.

**Referee comment:**
22. Line 444. Change to timing and strength of stratification.
**Author response:**
Thank you, we changed the text accordingly:
> L457: Our work demonstrates that oxygen dynamics in Lake Mendota are strongly governed by the stratification strength and timing in the water column.

**Referee comment:**
23. Line 469. Hopefully Chl-a will show the same results.
**Author response:**
Please see our reply to the referee's first major comment above.

**Referee comment:**
24. Line 504. Add But this does show a decadal shift in the extent of AF.
**Author response:**
Thank you, we revised the text using the suggestion by the referee:
> L518: For simplicity and due to limitations in Lake Mendota monitoring data post-2010, we focused the regression analysis of the Anoxic Factor in this study only on the pre-2010 period. The detection of this decadal shift in summer anoxia post-2010 highlights a hidden biological process that was not considered in the process-based model and may be due to an ecosystem shift in Lake Mendota that began in 2009, when the invasive spiny water flea (Bythothrephes longimanus) was detected in surprisingly high densities in the lake (Walsh et al., 2016b, 2018).

**Referee comment:**
25. Line 517. I really think the volume part of this model includes much of the physics associated with the volume of the hypolimnion and the length of stratification. This should be included. If you don't it really looks like this model gives a completely different interpretation.
**Author response:**
Please see our discussion of the deductive models' representation of physical processes at the beginning.

**Referee comment:**
26. Line 533. I really think you are being too hard on GLM. If you calibrated it better you should not have a consistent hypolimnetic bias. It has been shown to work well on many lakes, so I would not criticize it so hard. I really think the biggest problem was not calibrating the phytoplankton, by not doing that it affected many things. I think that is the number one thing for future model development. And the second thing would be trying to simulate the change in phytoplankton that occurred in 2010.
**Author response:**
We agree, and after investing seemingly years of our combined lives in modeling phytoplankton in Lake Mendota using GLM, we can say with certainty that it is very difficult.

We discuss several of these points in the manuscript, i.e. "improving the representation of phytoplankton and zooplankton dynamics in numerical models." (L575), "[…] numerical representations of phytoplankton life cycles (Hense, 2010; Shimoda and Arhonditsis, 2016), and/or allometric scaling (Shimoda et al., 2016) could significantly improve numerical phytoplankton predictions" (L579). Our statement regarding GLM-AED2's simulated discrepancies of hypolimnetic temperatures is rooted in a discussion of boundary conditions ("proximity of the atmospheric forcing boundary condition to the surface layers" (L556)) as well as the deep water mixing algorithm based on a vertical diffusivity approach instead of solving for turbulent diffusivity over the water column (like in a turbulence-closure scheme). Both points are essential part of GLM's design philosophy and should not be interpreted as critic. We agree that your neglection of a thorough phytoplankton calibration is an important shortcoming on our site that hopefully follow-up studies will focus on. We revised the text to reflect that: (a) shortcoming of calibration, and (b) we would need more data to even do a calibration:

> L573: Discrepancies between simulated and observed Anoxic Factors, therefore, could be rooted in our simplifications of the phytoplankton dynamics and its model parameter calibration, and the related organic matter fluxes, and highlight the importance of improving the representation of phytoplankton and zooplankton dynamics in numerical models. Simulating a magnitude of individual species rather than functional phytoplankton groups has been shown to improve numerical water quality and ecosystem predictions (Hellweger, 2017), though it is unclear if it could improve spring bloom predictions in Lake Mendota. This depends also on a more extensive monitoring program that measures and specifies specific phytoplankton species over the vertical gradient on a regular basis.

**Referee comment:**
27. Line 578. I don't see any reason why earlier stratification would cause as shallower thermocline. However, a warmer epilimnion could cause a shallower thermocline.
**Author response:**
This statement is grounded in Fig. 8c. We changed the text accordingly:

> L596: Further, a warmer epilimnion can cause the thermocline to become more shallow during the course of summer, which would cause the anoxia height to be spatially limited by a layer that is closer to the surface, hence more lake area would be anoxic. Increased oxygen depletion rates may also cause the anoxia height to be spatially limited by an earlier, and therefore lower, thermocline depth.

**Referee comment:**
28. Conclusions. Earlier you mention decadal shifts in the Abstract and Introduction. You found one using your models. This is a strength and should mention that by using GLM-AED you can say it was not driven by the physics, and it is probably driven by the changes in the biology.
**Author response:**
Thank you, we added a sentence to "5 Conclusions":

> L625: Further, our modelling framework detected a decadal shift in the Anoxic Factor starting in 2010, which was not driven by physical or chemical drivers, but probably related to an ecosystem shift caused by the invasive Bythothrephes longimanus.